# CONSTRUCTING CONFIDENCE INTERVALS FOR AVERAGE TREATMENT EFFECTS FROM MULTIPLE DATASETS

**Yuxin Wang, Maresa Schröder, Dennis Frauen, Jonas Schweisthal,**
**Konstantin Hess & Stefan Feuerriegel**
LMU Munich
Munich Center for Machine Learning (MCML)
`Yuxin.Wang1@lmu.de`

## ABSTRACT

Constructing confidence intervals (CIs) for the average treatment effect (ATE) from patient records is crucial to assess the effectiveness and safety of drugs. However, patient records typically come from different hospitals, thus raising the question of how multiple observational/experimental datasets can be effectively combined for this purpose. In our paper, we propose a new method that estimates the ATE from multiple observational/experimental datasets and provides valid CIs. Our method makes little assumptions about the observational datasets and is thus widely applicable in medical practice. The key idea of our method is that we leverage prediction-powered inferences and thereby essentially 'shrink' the CIs so that we offer more precise uncertainty quantification as compared to naïve approaches. We further prove the unbiasedness of our method and the validity of our CIs. We confirm our theoretical results through various numerical experiments.

## 1 INTRODUCTION

Estimating the *average treatment effect (ATE)* together with *confidence intervals (CIs)* is relevant in many fields, such as medicine, where the ATE is used to assess the effectiveness and safety of drugs (Glass et al., 2013; Feuerriegel et al., 2024). Nowadays, there is a growing interest in using observational datasets for this purpose, for example, electronic health records (EHRs) and clinical registries (Johnson et al., 2016; Corrigan-Curay et al., 2018; Hong, 2021). Importantly, such observational datasets typically originate from different hospitals, different health providers, or even different countries (Colnet et al., 2024), thus raising the question of *how to construct CIs for ATE estimation from multiple observational datasets*.

**Motivating example:** During the COVID-19 pandemic, the effectiveness and safety of potential drugs and vaccines were often assessed from electronic health records that originated from different hospitals to rapidly generate new evidence with treatment guidelines (Tacconelli et al., 2022). For example, one study (Wong et al., 2024) estimated the effect of *nirmatrelvir/ritonavir* (also known under the commercial name "paxlovid") in patients with COVID-19 diagnosis on 28-day all-cause hospitalizations from data obtained through a retrospective, multi-center study. The study eventually reported not only the ATE but also the corresponding CIs to allow for uncertainty quantification, which is standard in medicine (Kneib et al., 2023). However, the question of how one could combine the predictions and confidence intervals from multiple studies to provide better estimates remains.

Existing works for estimating ATEs from *multiple* datasets can be loosely categorized by **(a)** which datasets are used and **(b)** the underlying objective as follows (see Fig. 1): **(a)** The underlying patient data can come either from experimental datasets (i.e., randomized controlled trials; RCTs) and/or observational datasets (Feuerriegel et al., 2024). Both require tailored methods as the propensity score is known in RCTs but not in observational data and must thus be estimated (Rubin, 1974). We later focus on a setting where the ATE is estimated from multiple observational datasets, but we also provide an extension for combinations of RCT and observational datasets. **(b)** Much of the literature focused on estimating ATEs from multiple datasets focuses on *point estimates* (Kallus et al., 2018; John et al., 2019; Yang & Ding, 2020; Guo et al., 2022; Hatt et al., 2022; Demirel et al., 2024),

but ***not*** uncertainty quantification. However, valid CIs are needed in medicine to ensure reliable decision-making, because of which the existing methods are *not* applicable for medical applications.

**Our method:** In this paper, we propose a novel method to *construct valid CIs for ATE estimation from multiple observational datasets*. Specifically, we consider the following setting: we have one (potentially small) unbiased observational dataset $\mathcal{D}^1$ for which we assume unconfoundedness (i.e., all confounders observed) and another large-scale observational dataset $\mathcal{D}^2$ for which we allow for unobserved confounders. Then, the key idea of our method is that we tailor prediction-powered inference to our task so that we can essentially 'shrink' the CI and thus offer more precise uncertainty quantification as compared to a naïve approach. We further present an extension of our method where we 'shrink' CIs in settings with a combination of RCT and observational data.

*Why are naïve approaches precluded?* One may think that one can simply concatenate both datasets to compute a pooled ATE, yet this is not possible as the second dataset $\mathcal{D}^2$ may be confounded and, hence, the overall ATE estimate would be *biased*. A different, naïve approach is to simply construct finite-sample CIs from $\mathcal{D}^1$ to obtain an unbiased ATE with valid CIs. Yet, the additional power of the second dataset $\mathcal{D}^2$ (e.g., the information about the treatment assignment and thus the propensity score) would be ignored so that the CIs are *too conservative* (Aronow et al., 2021).

**Intuition behind our method:** Even though the second dataset $\mathcal{D}^2$ is large and employing it in addition to $\mathcal{D}^1$ may help shrink the estimation variance, it may be confounded, leading to biases in the downstream estimation. Therefore, using the second dataset $\mathcal{D}^2$ directly for inference could lead to biased CATE estimates. As a remedy, we derive a prediction-powered inference estimator where we decompose the variance of the population-level estimate of the CATE into two parts: one part comes from the estimation variance of the CATE on dataset $\mathcal{D}^2$, while the second part is due to the difference in estimators of ATEs across both datasets $\mathcal{D}^1$ and $\mathcal{D}^2$. The estimation variance of the first part can be significantly decreased with access to a large-scale dataset $\mathcal{D}^2$. Interestingly, the second part allows us to account for potential confounding bias in $\mathcal{D}^2$ and thus still derives valid CIs ($\rightarrow$ Theorem 4.2).[1]

Our **main contributions** are three-fold:[2] (1) We propose a new method to construct CIs for the ATE from multiple observational datasets. We further extend our method to combinations of RCTs and observational datasets. (2) We prove that our method is a consistent estimator and gives valid CIs. (3) We perform experiments with medical data to demonstrate the effectiveness of our method.

## 2 RELATED WORK

We give an overview of three literature streams relevant to our work: (i) methods for constructing CIs for the ATE that solely rely on a *single* dataset; (ii) methods that estimate the ATE from *multiple* datasets; and (iii) prediction-powered inference.

**Estimating CIs for the ATE:** Several works focus on constructing CIs for the ATE (Bang & Robins, 2005; van der Laan & Rubin, 2006). One literature stream addresses asymptotically normal data, which typically results in $\sqrt{n}$-consistent, asymptotically unbiased, normally distributed estimators. Another literature stream focuses on finite-sample settings, yet these works impose strong assumptions, such as that the data

| Methods | Dataset setting | Uncertainty quantification |
|---|---|---|
| Kallus et al. (2018), Hatt et al. (2022), Demiral et al. (2024) | RCT + Obs. | ✗ |
| Yang et al. (2020), Guo et al. (2021) | Obs. + Obs. | ✗ |
| van der Laan et al. (2024)[†] | RCT + Obs. | ✓ |
| ***Ours*** | Obs. + Obs. (RCT + Obs.) | ✓ |

[†] Constrained to ATE estimators based on TMLE, while we focus on arbitrary ATE estimators.

Figure 1: Key works aimed at ATE estimation from multiple datasets.

come from an RCT (Aronow et al., 2021) or assume unconfoundedness with relaxed overlap assumptions (Armstrong & Kolesár, 2021). However, this literature stream focuses on ATE estimation from a single dataset, which is unlike our work.

---

[1]We refer to Barnard (1949) and refer to a confidence interval as "valid" when the interval achieves its stated coverage probability. For example, a 95% confidence interval is valid if, under repeated sampling, it contains the true parameter value approximately 95% of the time. Validity ensures the interval accurately reflects the level of uncertainty about the estimate.

[2]Code and data are available via `https://github.com/Yuxin217/causalppi`

**ATE estimation from *multiple* datasets:**[3] Existing methods can be grouped by (a) the underlying dataset setting and (b) the objective, that is, whether the method provides point estimates or uncertainty quantification (see Fig. 1): (a) Some methods focus on settings with RCT + observational data (e.g, Kallus et al., 2018; Chen et al., 2021; Hatt et al., 2022; Demirel et al., 2024).[4] Other methods focus on multiple observational datasets (e.g., Yang & Ding, 2020; Guo et al., 2022), which is also our focus. The former is typically easier because the propensity score is known. In contrast, in the latter, the propensity score is unknown and must be estimated to account for the covariate shift across treated and non-treated patients. (b) Most works focus on only point estimation (e..g, Kallus et al., 2018; Chen et al., 2021; Hatt et al., 2022; Yang & Ding, 2020; Guo et al., 2022; Demirel et al., 2024), but **not** uncertainty quantification.

Closest to our method is the work by van der Laan et al. (2024). Yet, there are crucial *differences*: the method focuses on (i) RCT+observational data, (ii) has a different ATE estimation process that can lead to numerical instabilities, and (ii) has limited flexibility in how $\mathcal{D}^2$ is leveraged (e.g., the estimators for $\mathcal{D}^1$ and $\mathcal{D}^2$ must be identical, despite that different estimators may be beneficial due to the different size and nature of the datasets). We discuss the differences to our method in Appendix H.

**Prediction-powered inference (PPI):** Angelopoulos et al. (2023a;b) proposed the PPI framework for performing valid statistical inference from a given dataset when the dataset is supplemented with predictions from a machine-learning model (a brief overview is in Section C). Several extensions have been proposed recently (e.g., Fisch et al., 2024; Zrnic & Candès, 2024). So far, PPI was derived mostly for traditional statistical quantities (e.g., mean, median, quantile). For example, Demirel et al. (2024) propose a method based on PPI to generalize point estimates of causal effects from one population to a target population. However, they do not provide uncertainty quantification. To the best of our knowledge, there is **no** work that has tailored PPI to construct CIs for ATE estimation, which is our novelty.

## 3 PROBLEM SETUP

We consider the standard setting for ATE estimation from observational data (e.g., Imbens, 2004; Rubin, 2006; Shalit et al., 2017; Hatt et al., 2022), which we extend to multiple datasets.

**Setting:** We consider a setting with a small observational dataset $\mathcal{D}^1 = \{(x_i^1, a_i^1, y_i^1)\}_{i=1}^n$ and a large-scale observational dataset $\mathcal{D}^2 = \{(x_j^2, a_j^2, y_j^2)\}_{j=1}^{N'}$ (see left part in Fig. 2). We use a discrete variable $d \in D = \{1, 2\}$ to refer to the datasets and thus indicate from which dataset variables are (e.g., $X^d \in \mathcal{D}^d$ for $\mathcal{D}^d \in \{\mathcal{D}^1, \mathcal{D}^2\}$). We omit the superscripts for $x_i^d$ and use $x_i$ and $x_j$ (in color) instead to denote that patients belong to $\mathcal{D}^1$ and $\mathcal{D}^2$, respectively, in order to avoid misunderstandings with squared values. Furthermore, we use $n$ and $N'$ to denote the size of the datasets $\mathcal{D}^1$ and $\mathcal{D}^2$, respectively, with $n \ll N'$. Without loss of generality, it is straightforward to extend our method to more than two datasets simply by concatenating them into $\mathcal{D}^2$.

Both datasets have patient information about treatments, outcomes (e.g., tumor size, length of hospital stay), and covariates (e.g., the age or sex of a patient). Formally, both datasets consist of assigned treatments $a_i^d \in \mathcal{A} = \{0, 1\}$, outcomes $y_i^d \in \mathcal{Y} \subseteq \mathbb{R}$, and covariates $x_i^d \in \mathcal{X} \subseteq \mathbb{R}^q$ for dataset $d \in \{1, 2\}$ and with $i = 1 \ldots n$ (for $d = 1$) and $j = 1 \ldots N'$ (for $d = 2$). We denote random variables by capital letters $X^d$ with realizations $x^d$. Later, we use sample splitting, and let $N$ denote the sample size of one half of $\mathcal{D}^2$, i.e., $N = \frac{1}{2}N'$. Our setting is relevant for a variety of practical applications in medicine where electronic health records are collected from different environments, such as from different hospitals or different countries.

---

[3]Several methods have also aimed at estimating heterogeneous treatment effects (HTEs) from multiple datasets (e.g., Johansson et al., 2016; Schweisthal et al., 2024). However, estimating HTEs is more challenging than estimating the ATE because of the variation across subpopulations and the larger risk of overlap violations. Importantly, using HTEs for computing ATEs is suboptimal, which is well-established in efficiency theory for ATE estimation (Kennedy, 2016) and would lead to so-called plug-in bias (Curth & Van der Schaar, 2021). Hence, methods for HTE estimation are *orthogonal* to our work.

[4]There are further specialized settings, yet which are different from ours. For example, some works estimate long-term outcomes by combining RCT+observational data (Athey et al., 2020; Ghassami et al., 2022; Imbens et al., 2024). Even others aim to increase the efficiency of trial analyses (Schuler et al., 2022; Liao et al., 2023).

We assume that datasets are sampled i.i.d. from populations $(X, A, Y, D) \sim \mathbb{P}$ and $(X^d, A^d, Y^d) = (X, A, Y \mid D = d) \sim \mathbb{P}^d$ with $d \in \{1, 2\}$, but with the same marginal distribution of $X^d$, i.e., $\mathbb{P}_X^1 = \mathbb{P}_X^2$ and $X = X^1 = X^2$ in $\mathbb{P}$. We later also generalize our theory to settings with distribution shifts and finite populations in Appendix B.1 and Appendix B.2, respectively. Given that we focus on observational datasets, the treatment assignment rule may vary, and we thus define the dataset-specific propensity score via $\pi^d(x) = \mathbb{P}\left(A^d = 1 \mid X^d = x\right), d \in \{1, 2\}$. Formally, we assume that the propensity score may differ across the two datasets (i.e., $\pi^1 \neq \pi^2$). This is common in medical practice, where different hospitals or countries have different treatment guidelines.

**Target estimand:** We adopt the potential outcomes framework (Neyman, 1923; Rubin, 2005) to formalize our causal inference task. Let $Y^d(a) \in \mathcal{Y}$, $d \in \{1, 2\}$ denote the potential outcome in different distribution $\mathbb{P}^d$ for treatment intervention $A^d = a$. In this paper, we are interested in estimating the ATE on the target population (i.e., where the small dataset is sampled from), which is given by $\tau = \mathbb{E}\left[Y^1(1) - Y^1(0)\right]$, and then constructing corresponding CIs for $\tau$.

**Assumptions:** We make the following assumptions necessary for ATE identification and estimation. Of note, the following assumptions are standard in ATE estimation (Imbens, 2004; Rubin, 2005; Shalit et al., 2017). We distinguish the assumptions for the small dataset $\mathcal{D}^1$ and the large dataset $\mathcal{D}^2$.

**Assumption 3.1.** *For dataset $\mathcal{D}^1$, it holds: (i) (Consistency) $A^1 = a \Rightarrow Y^1 = Y^1(a)$; (ii) (Overlap) $0 < \pi^1(x) < 1, \forall x \in \mathcal{X}$; (iii) (Unconfoundedness) $Y^1(0), Y^1(1) \perp\!\!\!\perp A^1 \mid X^1$.*

**Assumption 3.2.** *For dataset $\mathcal{D}^2$, it holds: (i) (Consistency) $A^2 = a \Rightarrow Y^2 = Y^2(a)$; (ii) (Overlap) $0 < \pi^2(x) < 1, \forall x \in \mathcal{X}$.*

The above assumptions are the *standard* assumptions for estimating ATEs from observational data and are widely used for any underlying estimation method (Imbens, 2004; Rubin, 2006; Shalit et al., 2017). Consistency usually holds as long as health information is accurately and systematically recorded. Overlap can be ensured through preprocessing (e.g., clipping). Unconfoundedness is plausible in digital health settings due to the growing availability of rich electronic health records.[5]

The above assumptions are consistent with the literature studying multiple observational datasets (Yang & Ding, 2020; Guo et al., 2022). Note that the assumptions for dataset $\mathcal{D}^2$ are weaker as compared to dataset $\mathcal{D}^1$. ● For $\mathcal{D}^1$, we assume that there is no unobserved confounding, but the propensity score is unknown. This is often the case in specialized medical facilities where patients receive close supervision and where thus all critical health measurements are reported, which is typically the case in cancer care (Castellanos et al., 2024) and in intensive care units (Johnson et al., 2016). Needless to say, our assumption is still considerably weaker than assuming an RCT because we allow that the treatment assignment mechanism varies greatly across subpopulations, is unknown, and must thus be estimated. Nevertheless, RCTs are a special case of the setting considered in our general framework in which the propensity score $\pi^1$ is known. ● For $\mathcal{D}^2$, we do *not* make the latter assumption but instead allow for unobserved confounding. This is often the case when data are recorded by general practitioners where the need for documentation is typically not as strictly enforced as in other medical facilities.

$\Rightarrow$ In sum, $\mathcal{D}^1$ would naturally lead to *unbiased* ATE estimation but suffers from a *large estimation variance* due to the small sample size. In contrast, $\mathcal{D}^2$ has a larger size and thus *more statistical power* but could lead to *biased* estimates *due to unobserved confounding*.

## 4 OUR METHOD FOR ATE ESTIMATION FROM MULTIPLE DATASETS

**Overview:** The general idea of our approach is shown in Fig. 2. Ⓐ **Measure of fit:** We first use sample splitting to compute a measure of fit, $m_\theta$, to estimate the ATE on the large, observational dataset $\mathcal{D}^2$. Here, we use a state-of-the-art method based on the DR-learner (Wager, 2024). We refer to the estimate as $\hat{\tau}_2(x)$, where $\tau_2(x) = \mathbb{E}[Y^2(1) - Y^2(0) \mid X^2 = x]$. Yet, $\hat{\tau}_2(x)$ can be biased due to unobserved confounding, because of which we later need to adjust for this via the so-called rectifier (see below). Ⓑ **Influence function estimation:** We compute the non-centered

---

[5]Furthermore, advances in sensitivity analysis (Frauen et al., 2024; Oprescu et al., 2023) and partial identification (Duarte et al., 2024) offer complementary pathways to relax this assumption. The existing works from Fig. 1 for causal inference from multiple datasets generally make this assumption. In that sense, our work makes *weaker* assumptions that are more realistic as we allow for unobserved confounders in $\mathcal{D}^2$.

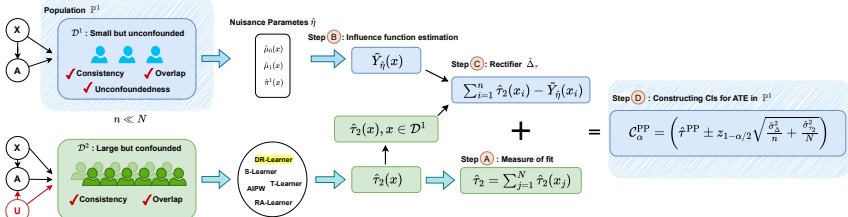

Figure 2: **Overview of our method.** To construct CIs for the ATE with two observational datasets from the same population but with different assumptions, we leverage prediction-powered inferences: we decompose our task into computing a measure of fit (i.e., estimating the ATE on the large dataset $\mathcal{D}^2$ via the DR-learner, given by $\hat{\tau}_2(x)$) and a rectifier $\hat{\Delta}_\tau$ (i.e., that measures the differences in ATE estimates across both datasets $\mathcal{D}^1$ and $\mathcal{D}^2$). However, finding a rectifier for our task is non-trivial and requires a careful derivation in order to ensure asymptotically valid CIs ($\rightarrow$ our Theorem 4.2).

influence function score $\tilde{Y}_{\hat{\eta}}(x)$ for the observational dataset $\mathcal{D}^1$. This is later used in the rectifier. Ⓒ **Rectifier:** We compute the rectifier $\Delta_\tau$, which we use to adjust for the bias between datasets $\mathcal{D}^1$ and $\mathcal{D}^2$. This allows us to transform the biased estimates $\hat{\tau}_2(x)$ into unbiased estimates of the ATE in population. Ⓓ **Constructing CIs:** Eventually, we compute the CIs $\mathcal{C}_\alpha^{\text{PP}}$ for significance level $\alpha$.

**Why is the above task challenging?** *First,* there is an information gap between the datasets $\mathcal{D}^1$ and $\mathcal{D}^2$. This means that different datasets come from different distributions, and we do not have any prior knowledge of the relationship between the datasets, except that the marginal distributions of $X^d$ are the same. In particular, there can be a distribution shift due to various reasons, such as unobserved effect modifiers (i.e., variables that change the treatment effect even if they are no confounders) and different treatment assignment mechanisms, because of which the propensity scores may be different across both datasets. We later propose a rectifier that accounts for such distribution shifts through an augmented inverse-propensity (AIPW) based estimation. Further, the propensity scores must be estimated, which introduces another source of uncertainty. *Second,* due to the fundamental problem of causal inference (Rubin, 1974), the ATEs are not directly observed but must be estimated while considering the aforementioned distribution shift. Further, such estimates must be asymptotically valid so that we can later derive CIs that are also asymptotically valid ($\rightarrow$ see our Theorem 4.2).

Below, we describe the steps Ⓐ–Ⓓ in detail. Pseuocode is in Algorithm 1.

**Step Ⓐ: Measure of fit.** The first step is to use sample splitting and estimate the conditional average treatment effect (CATE) $\tau_2(x) = \mathbb{E}[Y^2(1) - Y^2(0) \mid X^2 = x]$ on the half of the large-scale dataset $\mathcal{D}^2$ of size $N$ (after sample splitting). Let $\hat{\tau}_2(x)$ denote an arbitrary CATE estimator (which may be biased due to, e.g., unobserved confounding). For example, we could choose the DR-learner due to its fast convergence rate and several favorable theoretical properties (Kennedy, 2023). Needless to say, our method is also applicable to other estimators. Formally, we train $\hat{\tau}_2(x)$ and then yield the measure of fit on $\mathcal{D}^2$ via $\hat{\tau}_2 = \frac{1}{N} \sum_{j=1}^{N} \hat{\tau}_2(x_j)$ by sample splitting.

**Step Ⓑ: Influence function estimation.** For our proposed rectifier, we later need the non-centered influence function (IF) score on $\mathcal{D}^1$. Ideally, one would directly compute the difference in ATEs across both datasets for the rectifier, but this is impossible since the ATEs are not directly observed but rather need to be estimated. The estimation needs to be both valid and unbiased to later yield valid CIs.

We observe that the average of the non-centered IF score of the AIPW estimator is an unbiased estimation of ATE and is asymptotically normally distributed. This is beneficial for two reasons: (i) we get an unbiased estimate, which allows us to later obtain an unbiased ATE in population, and (ii) the estimate is asymptotically normal so that we later can derive valid CIs.

Formally, the non-centered IF score of the AIPW estimator (Robins & Rotnitzky, 1995) is given by

$$\tilde{Y}_{\hat{\eta}}(x_i) = \left( \frac{a_i^1}{\hat{\pi}^1(x_i)} - \frac{1 - a_i^1}{1 - \hat{\pi}^1(x_i)} \right) y_i^1 - \frac{a_i^1 - \hat{\pi}^1(x_i)}{\hat{\pi}^1(x_i) \left( 1 - \hat{\pi}^1(x_i) \right)} \left[ \left( 1 - \hat{\pi}^1(x_i) \right) \hat{\mu}_1(x_i) + \hat{\pi}^1(x_i) \hat{\mu}_0(x_i) \right],$$

$$\tag{1}$$

where the nuisance functions $\hat{\eta}(x) = \left( \hat{\mu}_0(x), \hat{\mu}_1(x), \hat{\pi}^1(x) \right)$ are plug-in estimators from $\mathcal{D}^1$, while $\hat{\mu}_a(x)$ are estimated response functions for $\mu_a(x) = \mathbb{E}[Y^1 \mid X^1 = x, A^1 = a]$ from $\mathcal{D}^1$. Then,

the AIPW estimator is $\hat{\tau}^{\text{AIPW}} = \frac{1}{n} \sum_{i=1}^{n} \tilde{Y}_{\hat{\eta}}(x_i)$. Leveraging results from causal inference literature (Wager, 2024), we have that $\hat{\tau}^{\text{AIPW}}$ is asymptotically normally distributed with $\sqrt{n}\left(\hat{\tau}^{\text{AIPW}} - \tau\right) \xrightarrow{\mathcal{N}} \left(0, V^{\text{AIPW}}\right)$, where

$$V^{\text{AIPW}} = \text{Var}\left[\mu_1(X) - \mu_0(X)\right] + \mathbb{E}\left[\left(A^1 \frac{Y^1 - \mu_1(X)}{\pi^1(X)}\right)^2\right] + \mathbb{E}\left[\left((1 - A^1)\frac{Y^1 - \mu_0(X)}{1 - \pi^1(X)}\right)^2\right].^{6}$$

Hence, the above procedure allows us to estimate the ATE for $\mathcal{D}^1$ and construct the corresponding CI with non-centered influence function scores, which we then use in the rectifier to assess the bias between both datasets $\mathcal{D}^1$ and $\mathcal{D}^2$.

**Step Ⓒ: Rectifier.** We now introduce our proposed rectifier to quantify the difference in ATE across both datasets. Formally, we define the rectifier $\Delta_\tau$ as the expectation of the difference between $\tilde{Y}_{\hat{\eta}}(x)$ and $\hat{\tau}_2(x)$ on $\mathcal{D}^1$. For individual observations $i$, we write $\hat{\Delta}_i = \tilde{Y}_{\hat{\eta}}(x_i) - \hat{\tau}_2(x_i)$. Note that our rectifier is carefully tailored to our task and is non-trivial because, due to the fundamental problem of causal inference (Rubin, 1974), the ATEs are never observed, but we need to leverage the influence functions score to be able to compute a valid and unbiased estimate. Formally, we have

$$\hat{\Delta}_\tau = \frac{1}{n} \sum_{i=1}^{n} \left[\tilde{Y}_{\hat{\eta}}(x_i) - \hat{\tau}_2(x_i)\right] = \frac{1}{n} \sum_{i=1}^{n} \left[\left(\frac{a_i^1}{\hat{\pi}^1(x_i)} - \frac{1 - a_i^1}{1 - \hat{\pi}^1(x_i)}\right) y_i^1 \right. \tag{2}$$
$$\left. - \frac{a_i^1 - \hat{\pi}^1(x_i)}{\hat{\pi}^1(x_i)\left(1 - \hat{\pi}^1(x_i)\right)} \left[\left(1 - \hat{\pi}^1(x_i)\right)\hat{\mu}_1(x_i) + \hat{\pi}^1(x_i)\hat{\mu}_0(x_i)\right] - \hat{\tau}_2(x_i)\right].$$

We now present Lemma 4.1, where we leverage results from the causal inference literature (Chernozhukov et al., 2018; Wager, 2024).

**Lemma 4.1** (follows from Wager (2024)). *Let $\mathcal{D}^1$ and $\mathcal{D}^2$ be sampled i.i.d under the assumptions above. Assume that we have estimated CATE estimator $\hat{\tau}_2(x)$ with sample splitting on $\mathcal{D}^2$, and have consistent estimated nuisance functions $\hat{\eta}(x) = \left(\hat{\mu}_0(x), \hat{\mu}_1(x), \hat{\pi}^1(x)\right)$ trained using cross-fitting with converge rate $\mathcal{O}(n^{-\alpha_\mu})$ and $\mathcal{O}(n^{-\alpha_\pi})$ on $\mathcal{D}^1$, i.e., $n^{-\alpha_\mu}(\hat{\mu}_a(x) - \mu_a(x)) \xrightarrow{p} 0$, $a = 0, 1$ and $n^{-\alpha_\pi}(1/\hat{\pi}^1(x) - 1/\pi^1(x)) \xrightarrow{p} 0$. Then, we have that*

$$\sqrt{n}\left(\hat{\Delta}_\tau - \tau + \mathbb{E}[\tau_2]\right) \to \mathcal{N}\left(0, \sigma_\Delta^2\right). \tag{3}$$

*Proof.* See Appendix A.1. $\qquad\square$

Then, the prediction-powered estimate of the ATE on $\mathcal{D}^1$ is computed via

$$\hat{\tau}^{\text{PP}} = \hat{\Delta}_\tau + \hat{\tau}_2 = \frac{1}{n} \sum_{i=1}^{n} \hat{\Delta}_i + \frac{1}{N} \sum_{j=1}^{N} \hat{\tau}_2(x_j) = \frac{1}{n} \sum_{i=1}^{n} \left[\tilde{Y}_{\hat{\eta}}(x_i) - \hat{\tau}_2(x_i)\right] + \frac{1}{N} \sum_{j=1}^{N} \hat{\tau}_2(x_j) \tag{4}$$

$$= \frac{1}{N} \sum_{j=1}^{N} \hat{\tau}_2(x_j) + \frac{1}{n} \sum_{i=1}^{n} \left[\left(\frac{a_i^1}{\hat{\pi}^1(x_i)} - \frac{1 - a_i^1}{1 - \hat{\pi}^1(x_i)}\right) y_i^1 \right. \tag{5}$$
$$\left. - \frac{a_i^1 - \hat{\pi}^1(x_i)}{\hat{\pi}^1(x_i)\left(1 - \hat{\pi}^1(x_i)\right)} \left[\left(1 - \hat{\pi}^1(x_i)\right)\hat{\mu}_1(x_i) + \hat{\pi}^1(x_i)\hat{\mu}_0(x_i)\right] - \hat{\tau}_2(x_i)\right].$$

**Step Ⓓ: Constructing CIs.** We now use the above PPI-based ATE estimate to construct our CIs. Let $\hat{\sigma}_{\tau_2}^2$ denote the empirical variance of $\hat{\tau}_2(x)$, and let $\hat{\sigma}_\Delta^2$ denote the empirical variance of $\hat{\Delta}_\tau$. Then, for significance level $\alpha \in (0, 1)$, our prediction-powered confidence interval is

$$\mathcal{C}_\alpha^{\text{PP}} = \left(\hat{\tau}^{\text{PP}} \pm z_{1-\frac{\alpha}{2}} \sqrt{\frac{\hat{\sigma}_\Delta^2}{n} + \frac{\hat{\sigma}_{\tau_2}^2}{N}}\right), \tag{6}$$

where $\hat{\sigma}_\Delta^2 = \frac{1}{n} \sum_{i=1}^{n} \left(\tilde{Y}_{\hat{\eta}}(x_i) - \hat{\tau}_2(x_i) - \hat{\Delta}_\tau\right)^2$, and $\hat{\sigma}_{\tau_2}^2 = \frac{1}{N} \sum_{j=1}^{N} \left(\hat{\tau}_2(x_j) - \hat{\tau}_2\right)^2$. We show theoretically that $\mathcal{C}_\alpha^{\text{PP}}$ is asymptotically valid in Theorem 4.2.

Equation 6 has several implications for how our method 'shrinks' CIs. (i) The width of the CIs depends on the size of the different datasets (which we later evaluate empirically as part of our

---

[6]Strong double robustness holds here due to the following reason. We use that the estimated nuisance functions are both consistent and that the RMSE of $\hat{\mu}_a(x)$ and $\hat{\pi}^1(x)$ decays fast enough. Then, the AIPW estimation is asymptotically normal around the oracle ATE.

sensitivity analyses). Hence, the width shrinks with a larger dataset $\mathcal{D}^1$ and/or a larger dataset $\mathcal{D}^2$. (ii) The width of the CIs depends on the estimation variance $\hat{\sigma}^2_{\tau_2}$ from the dataset $\mathcal{D}^2$. This is desired because our method is particularly designed for using large-scale but confounded datasets $\mathcal{D}^2$, so this term should shrink the CIs. (iii) The CIs further depend on the estimation variance of the rectifier $\hat{\sigma}^2_\Delta$. This term becomes smaller the less confounding the observational dataset $\mathcal{D}^2$ has.

**Theorem 4.2** (Validity of our prediction-powered CIs). *Let $\mathcal{D}^1$ and $\mathcal{D}^2$ are sampled i.i.d. under the assumptions from above. Further, assume that we have consistent and cross-fitted estimated nuisance functions $\hat{\eta}(x) = (\hat{\mu}_0(x), \hat{\mu}_1(x), \hat{\pi}^1(x))$ with converge rates $\mathcal{O}(n^{-\alpha_\mu})$, $\mathcal{O}(n^{-\alpha_\pi})$ on $\mathcal{D}^1$, i.e., $n^{-\alpha_\mu}(\hat{\mu}_a(x) - \mu_a(x)) \xrightarrow{p} 0$, $a = 0, 1$ and $n^{-\alpha_\pi}(1/\hat{\pi}^1(x) - 1/\pi^1(x)) \xrightarrow{p} 0$ and $\alpha_\mu + \alpha_\pi \geq 1/2$. We further assume a CATE estimator $\hat{\tau}_2(x)$ trained on $\mathcal{D}^2$. For some $p \in [0, 1]$, $\lim_{n,N\to\infty} \frac{n}{N} = p$. Fix $\alpha \in (0, 1)$, and let $\mathcal{C}^{PP}_\alpha = \left( \hat{\tau}^{PP} \pm z_{1-\alpha/2} \sqrt{\frac{\hat{\sigma}^2_\Delta}{n} + \frac{\hat{\sigma}^2_{\tau_2}}{N}} \right)$. Then, it holds that $\limsup_{n,N\to\infty} P(\tau \in \mathcal{C}^{PP}_\alpha) \geq 1 - \alpha$.*

*Proof.* See Appendix A.2 where we leverage Lemma 4.1. □

***Why is our method better than using the unconfounded dataset only?*** As shown in Equation 6, the width of our proposed CIs is mainly determined by the variance term $\sqrt{\frac{\hat{\sigma}^2_\Delta}{n} + \frac{\hat{\sigma}^2_{\tau_2}}{N}}$. When the $\hat{\tau}_2(x)$ is sufficiently accurate, the rectifier is almost equal to zero, i.e., $\hat{\Delta} \approx 0$. Then, the variance of the rectifier is significantly smaller than the variance of estimated non-centered IF scores, i.e., $\hat{\sigma}^2_\Delta \leq \hat{\sigma}^2_{\tau_2}$. Given the large size of $\mathcal{D}^2$, the variance of the estimated CATE goes to zero since the estimated variance should be divided by the sample size of $\mathcal{D}^2$, i.e., $N$ (after sample splitting). As a result, the variance (and thus the CI width) is smaller when using our method than when using only the unconfounded dataset, which means that our CIs are more narrow than such a naïve baseline.

The above theorem is crucial because it ensures that our PPI-based CIs are asymptotically valid. Further, note that the above theorem is our contribution: it does *not* directly follow from the PPI-based framework. Rather, we must carefully leverage theoretical guarantees for the estimand of interest and the chosen rectifier, which is one of our contributions.

## 5 EXTENSION OF OUR METHOD FOR RCT + OBSERVATIONAL DATASETS

We now extend our PPI-based method to combinations of RCT+observational data. Using an RCT dataset is a special case of $\mathcal{D}^1$. As a result, the propensity score is known, which simplifies the underlying task. Yet, the information gap between the datasets remains in that both come from different distributions (e.g., different populations $\mathcal{X}$, different effect modifiers, etc.).

A straightforward way to extend our method would be to apply our AIPW-based method directly with the known propensities. However, this may have disadvantages as it still requires the estimation of nuisance functions (response functions). Below, we describe the alternative method based on the inverse-propensity weighting (IPW) estimator, which makes necessary changes for the steps Ⓐ–Ⓓ. Note that we no longer need the step estimating the influence functions because the propensity score is known, meaning that we can directly estimate $\tau_1$ via the IPW estimator.

**Step Ⓐ: Measure of fit.** First, we compute the CATEs analogous to the above approach by using sample splitting to train $\hat{\tau}_2$ on $\mathcal{D}^2$. We thus yield the measure of fit, i.e., $\hat{\tau}_2 = \frac{1}{N} \sum_{j=1}^{N} \hat{\tau}_2(x_j)$.

**Step Ⓑ: IPW estimator.** Given the RCT dataset $\mathcal{D}^1$ and known propensity scores, we can compute the inverse-propensity weighted estimation of ATE via

$$\hat{\tau}_1 = \frac{1}{n} \sum_{i=1}^{n} \tilde{Y}_{\pi^1}(x_i) = \frac{1}{n} \sum_{i=1}^{n} \left( \frac{a_i^1 y_i^1}{\pi^1(x_i)} - \frac{(1 - a_i^1) y_i^1}{\pi^1(x_i)} \right), \tag{7}$$

which we later use in the rectifier (instead of the influence function score as in our method for multiple observational datasets).

**Remark 5.1.** *Let $\hat{\tau}_1$ denote the IPW estimator for the ATE of the RCT dataset ($\mathcal{D}^1$), $\hat{\tau}_1$ is asymptotically normally distributed, i.e., $\sqrt{n}(\hat{\tau}_1 - \tau) \xrightarrow{d} \mathcal{N}(0, \hat{\sigma}^2_1)$, where $\hat{\tau}_1 = \frac{1}{n} \sum_{i=1}^{n} \tilde{Y}_{\pi^1}(x_i)$, $\hat{\sigma}^2_1 = \frac{1}{n} \sum_{i=1}^{n} \left( \tilde{Y}_{\pi^1}(x_i) - \hat{\tau}_1 \right)^2$.*[7]

---

[7] The proof is standard and follows from, e.g., Wager (2024).

The above Remark 5.1 ensures that the estimate is asymptotically normal, which allows us to later obtain valid CIs.

**Step $\textcircled{c}$: Rectifier.** We now introduce our rectifier $\Delta_\tau$, which denotes the average difference between $\tilde{Y}_{\pi^1}(x)$ and $\hat{\tau}_2(x)$ on $\mathcal{D}^1$. Here, $\hat{\tau}_2(x)$ is trained CATE estimators on $\mathcal{D}^2$ based on sample splitting. Our rectifier then given by $\Delta_\tau = \mathbb{E}\left[\tilde{Y}_{\pi^1}(x) - \tau_2(x)\right]$.

Then, the prediction-powered estimate of the ATE on $\mathcal{D}^1$ is computed via

$$\hat{\tau}^{\text{PP}} = \frac{1}{N}\sum_{j=1}^{N}\hat{\tau}_2(x_j) + \frac{1}{n}\sum_{i=1}^{n}\hat{\Delta}_i = \frac{1}{N}\sum_{j=1}^{N}\hat{\tau}_2(x_j) + \frac{1}{n}\sum_{i=1}^{n}\tilde{Y}_{\pi^1}(x_i) - \hat{\tau}_2(x_i). \tag{8}$$

**Step $\textcircled{d}$: Constructing CIs.** We now use the above PPI-based ATE estimate to construct our CIs. Let $\hat{\sigma}^2_{\tau_2}$ denote the empirical variance of $\hat{\tau}_2(\mathbf{x})$, and let $\hat{\sigma}^2_\Delta$ denotes empirical variance of rectifier. Then, for significance level $\alpha \in (0,1)$, our prediction-powered CI is given by

$$\mathcal{C}^{\text{PP}}_\alpha = \left(\hat{\tau}^{\text{PP}} \pm z_{1-\frac{\alpha}{2}}\sqrt{\frac{\hat{\sigma}^2_\Delta}{n} + \frac{\hat{\sigma}^2_{\tau_2}}{N}}\right), \tag{9}$$

where $\hat{\Delta}_\tau = \frac{1}{n}\sum_{i=1}^{n}\hat{\Delta}_i$, $\hat{\sigma}^2_\Delta = \frac{1}{n}\sum_{i=1}^{n}\left(\hat{\Delta}_i - \hat{\Delta}_\tau\right)$, and $\hat{\tau}_2 = \frac{1}{N}\sum_{j=1}^{N}\hat{\tau}_2(x_j)$, $\hat{\sigma}^2_{\tau_2} = \frac{1}{N}\sum_{j=1}^{N}\left(\hat{\tau}_2(x_j) - \hat{\tau}_2\right)$. The following theorem shows that our above prediction-powered CI is valid.

**Theorem 5.2** (Validity of our prediction-powered CIs in RCT+observational setting). *Let $\mathcal{D}^1$ and $\mathcal{D}^2$ are sampled i.i.d. under the assumptions from above, and $\lim_{n,N\to\infty}\frac{n}{N} = p$ for some $p \in [0,1]$. Then, the prediction-powered confidence interval has valid coverage:* $\liminf_{n,N\to\infty} P\left(\tau \in \mathcal{C}^{\text{PP}}_\alpha\right) \geq 1-\alpha$.

*Proof.* See Appendix A.3 where we leverage Remark 5.1. $\square$

## 6 EXPERIMENTS

We now evaluate the effectiveness of our proposed method by examining the faithfulness and width of the constructed CIs. To this end, we follow prior research and perform experiments with both synthetic and medical data (e.g., Schröder et al., 2024; Schweisthal et al., 2024). Synthetic data has the advantage that we have access to the ground-truth CATEs and thereby can make comparisons against oracle estimates. Further, medical data allows us to demonstrate both the applicability and relevance of our method in practice.

### 6.1 SYNTHETIC DATA

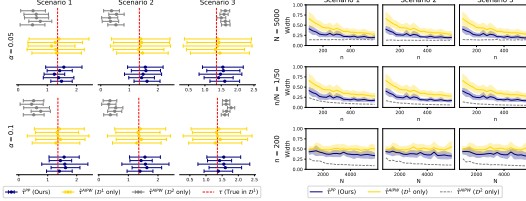

Figure 3: **Performance for synthetic data.** *Left:* We show the estimated CIs for five random seeds. The red line is the oracle ATE. Ideally, the CIs should be narrow but still overlap with the oracle ATE. *Right:* Shows in the width of the CIs averaged over five different seeds ($\alpha = 0.05$). Here, we vary the size of the different datasets given by $n$ ($\mathcal{D}^1$) and $N$ ($\mathcal{D}^2$). Note that $\hat{\tau}^{\text{AIPW}}$ ($\mathcal{D}^2$ only) is shown in intentionally shown in gray: it is *not* faithful as seen in the left plot and therefore *not* a valid baseline. $\Rightarrow$ Our method yields faithful CIs, and CIs are shorter as desired.

**Data:** Inspired by Demirel et al. (2024), we simulate samples from a data-generating process with a confounder $X^d \in [-1,1]$ and an unobserved confounder $U^2 \in \mathcal{U} \subseteq [-1,1]$, a binary treatment $A^d \in \{0,1\}$, and a real-valued outcome $Y^d \in \mathbb{R}$ for $d \in \{1,2\}$. We generate the potential outcomes $Y^d(a)$, $d \in \{1,2\}$ conditioned on $X^d = x$ and $U^d = u$ by sampling from a Gaussian process $\mathcal{GP} : [-1,1]^2 \to \mathbb{R}$ with mean function $m(x,u) = 0$ and kernel function $k\left((x,u),(x',u')\right)$. We choose a composite kernel by adding a squared-exponential (SE) kernel to model the *local* variation and a linear kernel to model trends in the outcome. We thus have $k\left((x,u),(x',u')\right) = \alpha_x xx' + \alpha_u uu' + \exp\left[-\frac{(x-x')}{2l_x^2} - \frac{(u-u')}{2l_u^2}\right]$, with configuration parameters $\theta = \{\alpha_x, \alpha_u, l_x, l_u\} \in \mathbb{R}^4_+$. We can simulate different confounding strengths by varying the value of $\theta$.

We generate the covariates of observational datasets $\mathcal{D}^1$ and $\mathcal{D}^2$ by sampling $X^d, U^2 \sim \text{Uniform}[-1,1]$ for $d \in \{1,2\}$ independently. For each patient, treatments assignments are sampled via $A_i \sim \text{Bernoulli}(P(A=1 \mid x_i,u_i))$, where the probability of treatment assignment is

generated similarly to Equation 6.1 via a logit function $L_\pi(x, u)$ sampled from $\mathcal{GP}_{\theta_\pi}(x, u)$, where $\theta_\pi = \{\alpha_x^\pi, \alpha_u^\pi, l_x^\pi, l_u^\pi\}$. A larger value of $\alpha_u$ and a smaller value of $l_u$ implies stronger confounding. The observed outcomes are computed via $Y = (1 - A) \cdot \mathcal{GP}_{\theta_0}(x, u) + A \cdot \mathcal{GP}_{\theta_1}(x, u)$.

We generate $n = 200$ ($\mathcal{D}^1$) and $N = 5000$ ($\mathcal{D}^2$) samples. For $\mathcal{D}^1$, we set $\alpha_u = 0$ and $l_u = 10^6$ to prevent confounding. For $\mathcal{D}^2$, we use different values of $\theta$ to generate different confounding scenarios: • **Scenario 1:** little confounding ($\alpha_u = 0$, $l_u = 10^6$). • **Scenario 2:** medium confounding ($\alpha_u = 0$, $l_u = 0.5$). • **Scenario 3:** heavy confounding ($\alpha_u = 10$, $l_u = 0.5$). Further details and illustrations about the data-generation process are given in Appendix F.1. Altogether, we generate over 60 different datasets under varying configurations for the evaluations below.

**Baselines:** We compare our PPI-based method $\hat{\tau}^{PP}$ for constructing CIs against the following baseline: **(1)** we estimate the ATE via the AIPW estimator $\hat{\tau}^{AIPW}$ only on the small dataset, named $\hat{\tau}^{AIPW}$ ($\mathcal{D}^1$ only) which is the **naïve baseline**; **(2)** we estimate the ATE via the AIPW estimator on the large, confounded dataset, named $\hat{\tau}^{AIPW}$ ($\mathcal{D}^2$ only); and **(3)** we report the oracle value for $\tau$ in $\mathcal{D}^1$.

**Main results:** Fig. 3 (left). We observe the following: **(1)** The CIs from our method overlap with the oracle ATE (in red), which shows that our method is faithful. **(2)** In contrast, the baseline $\hat{\tau}^{AIPW}$ ($\mathcal{D}^2$ only) rarely covers the oracle ATE and is thus *not* faithful. This can be expected since the dataset computes the CIs based on the confounded dataset and, hence, yields biased estimates. The unfaithfulness becomes especially evident in Scenario 3 where data under large confounding is generated. **(3)** Our method generates CIs that are more narrow as compared to the naïve baseline $\hat{\tau}^{AIPW}$ ($\mathcal{D}^1$ only) . For example, in the left plot, our CIs are, on average, smaller by 49.99% (Scenario 1), 55.37% (Scenario 2), and 55.35% (Scenario 3). ⇒ *Takeaway: Our PPI-based method yields faithful CIs but where the width of the CIs is clearly shorter.* Hence, our method performs the best.

**Sensitivity to dataset size:** Fig. 3 (right) compares the sensitivity across different dataset sizes. **(1)** Our method generates CIs that are again more narrow and, therefore, superior. We observe that our method performs better in terms of widths of CIs than the naïve baseline. **(2)** The advantages of our method become pronounced for setting where $N \gg n$ as expected (see right plot, top row).

## 6.2 MEDICAL DATA

**Dataset:** We now provide a case study that demonstrates the applicability of our method to medical datasets. We chose two common datasets: the **MIMIC-III** dataset (Johnson et al., 2016) and a Brazilian **COVID-19** dataset (Baqui et al., 2020). • **MIMIC-III** contains health records from patients admitted to intensive care units at large hospitals. We aim to estimate the average red blood cell count of all patients after being treated with mechanical ventilation. Our estimation is based on 8 confounders from medical practice (e.g., respiratory rate, hematocrit). • The **COVID-19** dataset contains health records of hospitalizations in Brazil across different regions and from patients with different socio-economic backgrounds. We are interested in predicting the effect of comorbidities on the mortality of COVID-19 patients. We created two different splits of the original dataset into $\mathcal{D}^1$ and $\mathcal{D}^2$: (i) we split by regions of the hospitals in Brazil (i.e., North and Central-South) and (ii) by ethnicity of participants (i.e., White and others). Further details are in Appendix F.2.

Table 1: **Results for different medical datasets.** We report the RMSE of the ATE estimator and the width of the CIs. The results for $\hat{\tau}^{AIPW}$ ($\mathcal{D}^2$ only) are shown in gray because the estimator is *not* faithful and therefore also *not* a viable baseline. Reported is the average performance over 5 random seeds.

| Dataset | MIMIC-III | | COVID-19 (by region) | | COVID-19 (by ethnicity) | |
|---|---|---|---|---|---|---|
| | RMSE | Width | RMSE | Width | RMSE | Width |
| $\hat{\tau}^{AIPW}$ ($\mathcal{D}^1$ only) | **0.057** | 0.077 | 7.591 | 8.479 | 39.970 | 0.081 |
| $\hat{\tau}^{AIPW}$ ($\mathcal{D}^2$ only) | 0.058 | 0.003 | 17.125 | 0.311 | 39.999 | 0.004 |
| $\hat{\tau}^{PP}$ (Ours) | 0.057 | 0.023 | **7.131** | 2.341 | **39.968** | 0.026 |

Smaller is better. Best value in bold.

**Results:** The results are in Table 1. We again compare the CIs of our estimator against the baselines from above. We further report the root means squared error for the factual outcomes. We find: **(1)** Our method achieves the smallest RMSE, which indicates that underlying patterns in the data are well captured. **(2)** Our method obtains the smallest yet valid CIs. Compared to $\hat{\tau}^{AIPW}$ ($\mathcal{D}^1$ only), our methods leads a ∼3.5x reduction in the width of CIs. **(3)** $\hat{\tau}^{AIPW}$ ($\mathcal{D}^2$ only) is known to be biased. This explains why the RMSE is sometimes considerably larger than the RMSE for the other methods, which again corroborates our findings that $\hat{\tau}^{AIPW}$ is not faithful. ⇒ *Takeaway: Our PPI-based method is effective for medical data.*

## 6.3 RESULTS FOR RCT+OBSERVATIONAL DATA

**Data:** We use the same data-generating process from above. However, we now mimic an RCT for $\mathcal{D}^1$ by setting the unobserved confounder $U$ to zero and corresponding $\alpha = 0$ and $l_u = 10^6$. Details are in Appendix F.1.

**Baselines:** We now report our method based on IPW (instead of AIPW). We additionally implement the method by van der Laan et al. (2024), which allows to estimate the ATE from both datasets. We refer to this method by $\hat{\tau}^{\text{ATMLE}}$. However, we note that the method is often unstable: their method involves a matrix inversion, yet where the matrix is often singular, so that no CIs can be computed (see Appendix H for a detailed explanation). This later explains the fairly noisy performance of the baseline. For a fair comparison, we simply set the output in these cases to $\mathcal{D}^1$.

**Results:** Fig. 4 (left) shows the results. We find: **(1)** The CIs from our method cover the oracle ATE (in red), which shows that our method is faithful for the RCT+observational dataset. **(2)** $\hat{\tau}^{\text{ATMLE}}$ is faithful in the settings with little and medium confounding (Scenarios 2 and 3), but it fails in Scenario 3 where it is *not* faithful. **(3)** Our method generates CIs that are consistently more narrow compared to the baselines. ⇒ ***Takeaway: Our method performs best.***

**Sensitivity to dataset size**: In Fig. 4 (right), we analyze the role of dataset sizes. The results confirm our findings from above: Compared to $\hat{\tau}^{\text{ATMLE}}$, our method is much more stable. Further, our method generates CIs that are consistently more narrow and thus superior.

## 6.4 ADDITIONAL EXPERIMENTS

We provide further experiments to corroborate our above takeaways in (Appendix. G).

• **Variations of our method:** (1) We performed experiments where we instantiated our method using **neural networks** as regression models for estimating nuisance parameters in AIPW to offer more flexibility in learning representations of the covariate space (see Appendix G.1). (2) We performed experiments with **XGBoost** to show the applicability of our method to underlying base models

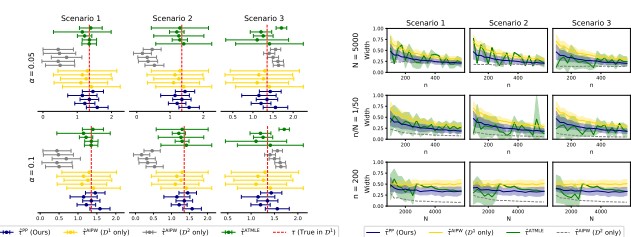

Figure 4: **Performance for synthetic data.** *Left:* We show the estimated CIs for five different seeds in RCT and observational datasets. *Right:* We show the width of the CIs averaged over five different seeds ($\alpha = 0.05$). ⇒ Our method is both stable and leads to CIs that are faithful and narrow, as desired.

for estimating nuisance parameters in AIPW (see Appendix G.2). • **Different settings:** (3) We varied the size of the covariate space to demonstrate the effectiveness of our method in settings with a **high-dimensional covariate space** (see Appendix G.3). (4) We varied the **covariate dependence** by increasing the collinearity in the input space (see Appendix G.4). (5) We varied the **strength of confounding** in $\mathcal{D}^1$ (see Appendix G.5). We found that our method performs best across all settings. (6) Oftentimes, estimates of treatment effects in RCT settings benefit when the propensity score is estimated (Su et al., 2023; Cai & van der Laan, 2018). Motivated by this, we applied our AIPW-based method to combinations of RCT and observational datasets (see Appendix G.6). Here, we find that we can improve the CI width further using our AIPW-based method. (7) We performed a refutation check in which we applied A-TMLE to combinations of two observational datasets (see Appendix G.7), but remind that this violates the assumptions of A-TMLE. As expected, A-TMLE underperforms, and our method remains clearly superior. (8) We expanded the sample size of $\mathcal{D}^1$ from 100 to 2500 to further assess the role of the size of $\mathcal{D}^1$ and the robustness of our method regarding the size of $\mathcal{D}^1$. We found that our method shows a clear margin and the results are as expected (see Appendix G.8).

## 7 DISCUSSION

**Relevance:** In this paper, we developed a new method for ATE estimation from multiple observational datasets. Our method is highly relevant to medical practice as it helps assess the effectiveness and safety of drugs. To this end, we perform rigorous uncertainty quantification by deriving and reporting valid CIs. **Limitations & future work:** One improvement is extending our method to other estimands like the CATE or to causal survival analysis. Future research may combine our method with pre-trained large language models (LLMs) or develop tailored neural network architectures on top of our method for text-based representations. However, as with any method of causal inference, the assumptions must be carefully assessed to ensure safe and reliable use.

**Acknowledgments.** Our research was supported by the DAAD program Konrad Zuse Schools of Excellence in Artificial Intelligence, sponsored by the Federal Ministry of Education and Research.

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

# A PROOFS

## A.1 SUPPORTING LEMMA

Below, we restate Lemma 4.1 in a more detailed way and provide the proof.

**Lemma 4.1** Let $\mathcal{D}^1$ and $\mathcal{D}^2$ be sampled i.i.d. under the assumptions above. We assume sample splitting; i.e., let $\mathcal{D}^1$ be split into $\mathcal{D}^{1,1}$ and $\mathcal{D}^{1,2}$, and let $\mathcal{D}^2$ be split into into $\mathcal{D}^{2,1}$ and $\mathcal{D}^{2,2}$. Let us assume that we have an estimated CATE estimator $\hat{\tau}_2(x)$ on $\mathcal{D}^{2,1}$. Further, we assume that we have estimated nuisance functions $\hat{\eta}^1(x) = \left(\hat{\mu}_0^1(x), \hat{\mu}_1^1(x), \hat{\pi}^{1,1}(x)\right)$ and $\hat{\eta}^2(x) = \left(\hat{\mu}_0^2(x), \hat{\mu}_1^2(x), \hat{\pi}^{1,2}(x)\right)$ that were estimated separately on $\mathcal{D}^{1,1}$ and $\mathcal{D}^{1,2}$ with converge rates

$$n^{-\alpha_\mu} \frac{1}{|\mathcal{D}^{1,1}|} (\hat{\mu}_a^1(x) - \mu_a^1(x)) \xrightarrow{p} 0, \qquad a = 0, 1,$$
$$n^{-\alpha_\pi} \frac{1}{|\mathcal{D}^{1,1}|} (1/\hat{\pi}^{1,1}(x) - 1/\pi^{1,1}(x)) \xrightarrow{p} 0, \tag{10}$$

for some constants with $\alpha_\mu$, $\alpha_\pi \geq 0$ and $\alpha_\mu + \alpha_\pi \geq 1/2$. We assume that Equation 10 also holds when the roles of $\mathcal{D}^{1,1}$ and $\mathcal{D}^{1,2}$ swapped. Then, we construct the rectifier on $\mathcal{D}^1$ given by

$$\hat{\Delta}_\tau = \hat{\tau}_{\text{AIPW}} - \hat{\tau}_2^{\mathcal{D}^1}, \tag{11}$$

where the cross-fitted estimator is

$$\hat{\tau}_{\text{AIPW}} = \frac{|\mathcal{D}^{1,1}|}{n} \hat{\tau}^{\mathcal{D}^{1,1}} + \frac{|\mathcal{D}^{1,2}|}{n} \hat{\tau}^{\mathcal{D}^{1,2}}, \tag{12}$$

$$\hat{\tau}^{\mathcal{D}^{1,1}} = \frac{1}{|\mathcal{D}^{1,1}|} \sum_{i \in \mathcal{D}^{1,1}} \tilde{Y}_{\hat{\eta}^2}(x_i) \tag{13}$$

$$= \frac{1}{|\mathcal{D}^{1,1}|} \sum_{i \in \mathcal{D}^{1,1}} \left( \hat{\mu}_1^2(x_i) - \hat{\mu}_0^2(x_i) + a_i \frac{y_i - \hat{\mu}_1^2(x_i)}{\hat{\pi}^{1,2}(x_i)} - (1 - a_i) \frac{y_i - \hat{\mu}_0^2(x_i)}{1 - \hat{\pi}^{1,2}(x_i)} \right),$$

$$\hat{\tau}^{\mathcal{D}^{1,2}} = \frac{1}{|\mathcal{D}^{1,2}|} \sum_{i \in \mathcal{D}^{1,2}} \tilde{Y}_{\hat{\eta}^1}(x_i) \tag{14}$$

$$= \frac{1}{|\mathcal{D}^{1,2}|} \sum_{i \in \mathcal{D}^{1,2}} \left( \hat{\mu}_1^1(x_i) - \hat{\mu}_0^1(x_i) + a_i \frac{y_i - \hat{\mu}_1^1(x_i)}{\hat{\pi}^{1,1}(x_i)} - (1 - a_i) \frac{y_i - \hat{\mu}_0^1(x_i)}{1 - \hat{\pi}^{1,1}(x_i)} \right),$$

and the evaluated CATE estimator on $\mathcal{D}^1$ is

$$\hat{\tau}_2^{\mathcal{D}^1} = \frac{1}{n} \sum_{i \in \mathcal{D}^1} \hat{\tau}_2(x_i). \tag{15}$$

All $x_i$ in the above and following equations are $x_i^1$, so we ignore the superscript for simplicity and use colors to distinguish. Then, we have that

$$\sqrt{n} \left( \hat{\Delta}_\tau - \tau + \mathbb{E}[\tau_2] \right) \Rightarrow \mathcal{N}\left(0, \sigma_\Delta^2\right). \tag{16}$$

*Proof.* To prove Equation 16, we show it in three steps: (1) decomposition of the rectifier $\hat{\Delta}_\tau$; (2) the central limited theorem for oracle nuisance functions and CATE estimator; and (3) error due to nuisance estimation. These are below. We use the '*' to refer to oracle estimates.

(1) **Decomposition of $\hat{\Delta}_\tau$.** First, let us define a new auxiliary random variable $Z_i = \tilde{Y}_{\hat{\eta}}(x_i) - \hat{\tau}_2(x_i)$. We then separate the rectifier $\hat{\Delta}_\tau$ (i.e., the average of $Z_i$) into two parts:

$$\hat{\Delta}_\tau = \frac{1}{n} \sum_{i=1}^n Z_i = \underbrace{\frac{1}{n} \sum_{i \in \mathcal{D}^1} \left[ \tilde{Y}_{\hat{\eta}}(x_i) - Y_\eta(x_i) \right]}_{\text{error due to nuisance estimation}} + \underbrace{\frac{1}{n} \sum_{i \in \mathcal{D}^1} \left[ Y_\eta(x_i) - \hat{\tau}_2(x_i) \right]}_{\text{oracle nuisance functions}}. \tag{17}$$

The first term in Equation 17 denotes the error introduced by using estimated nuisance functions, and the second term in Equation 17 denotes the mean of the differences between pseudo-outcomes based on the oracle nuisance function and the CATE estimator.

(2) **The central limited theorem for oracle nuisance functions and CATE estimator.** Note that, for the second term in Equation 17, $Y_\eta(x_i)$ are pseudo-outcomes based on oracle nuisance functions (with *no* dependency on any estimated functions on $\mathcal{D}^1$ or $\mathcal{D}^2$), while $\hat{\tau}_2(\cdot)$ is estimated on $\mathcal{D}^{2,1}$ (particularly, it is independent of $\mathcal{D}^1$). Thus, $Y_\eta(x)$ and $\hat{\tau}_2(x)$ are independent functions, which means that $Y_\eta(x_i) - \hat{\tau}_2(x_i)$ **are i.i.d. random variables**. Then, following the central limit theorem (CLT), we have

$$\sqrt{n}\left(\frac{1}{n}\sum_{i\in\mathcal{D}^1}[Y_\eta(x_i)-\hat{\tau}_2(x_i)]-\tau+\mathbb{E}[\tau_2]\right)\Rightarrow\mathcal{N}(0,\sigma_\Delta^2). \tag{18}$$

(3) **Error due to nuisance estimation.** We follow the proof in Wager (2024) to show that the estimation error is negligible. First, we rewrite the first term in Equation 17 as

$$\frac{1}{n}\sum_{i\in\mathcal{D}^1}\left[\tilde{Y}_{\hat{\eta}}(x_i)-Y_\eta(x_i)\right]=\frac{1}{n}\sum_{i\in\mathcal{D}^1}\tilde{Y}_{\hat{\eta}}(x_i)-\frac{1}{n}\sum_{i\in\mathcal{D}^1}Y_\eta(x_i)=\hat{\tau}_{\text{AIPW}}-\hat{\tau}_{\text{AIPW}}^*. \tag{19}$$

Second, following Equation 12, we can rewrite the oracle estimation as

$$\begin{aligned}\hat{\tau}_{\text{AIPW}}^*&=\frac{|\mathcal{D}^{1,1}|}{n}\hat{\tau}^{\mathcal{D}^{1,1},*}+\frac{|\mathcal{D}^{1,2}|}{n}\hat{\tau}^{\mathcal{D}^{1,2},*}\\&=\frac{|\mathcal{D}^{1,1}|}{n}\cdot\frac{1}{|\mathcal{D}^{1,1}|}\sum_{i\in\mathcal{D}^{1,1}}Y_\eta(x_i)+\frac{|\mathcal{D}^{1,2}|}{n}\cdot\frac{1}{|\mathcal{D}^{1,2}|}\sum_{i\in\mathcal{D}^{1,2}}Y_\eta(x_i).\end{aligned} \tag{20}$$

Moreover, we can decompose $\hat{\tau}^{\mathcal{D}^{1,1}}$ and $\hat{\tau}^{\mathcal{D}^{1,1},*}$ as

$$\begin{aligned}\hat{\tau}^{\mathcal{D}^{1,1}}&=\hat{m}_1^{\mathcal{D}^{1,1}}-\hat{m}_0^{\mathcal{D}^{1,1}},\\\hat{\tau}^{\mathcal{D}^{1,1},*}&=\hat{m}_1^{\mathcal{D}^{1,1},*}-\hat{m}_0^{\mathcal{D}^{1,1},*}\end{aligned} \tag{21}$$

where

$$\begin{aligned}\hat{m}_1^{\mathcal{D}^{1,1}}&=\frac{1}{|\mathcal{D}^{1,1}|}\sum_{i\in\mathcal{D}^{1,1}}\left(\hat{\mu}_1^2(x_i)+a_i\frac{y_i-\hat{\mu}_1^2(x_i)}{\hat{\pi}^{1,2}(x_i)}\right),\\\hat{m}_0^{\mathcal{D}^{1,1}}&=\frac{1}{|\mathcal{D}^{1,1}|}\sum_{i\in\mathcal{D}^{1,1}}\left(\hat{\mu}_0^2(x_i)+(1-a_i)\frac{y_i-\hat{\mu}_0^2(x_i)}{1-\hat{\pi}^{1,2}(x_i)}\right),\\\hat{m}_1^{\mathcal{D}^{1,1},*}&=\frac{1}{|\mathcal{D}^{1,1}|}\sum_{i\in\mathcal{D}^{1,1}}\left(\mu_1(x_i)+a_i\frac{y_i-\mu_1(x_i)}{\pi^{1,2}(x_i)}\right),\\\hat{m}_0^{\mathcal{D}^{1,1},*}&=\frac{1}{|\mathcal{D}^{1,1}|}\sum_{i\in\mathcal{D}^{1,1}}\left(\mu_0(x_i)+(1-a_i)\frac{y_i-\mu_0(x_i)}{1-\pi^{1,2}(x_i)}\right).\end{aligned} \tag{22}$$

Then, we have

$$\begin{aligned}&\hat{\tau}_{\text{AIPW}}-\hat{\tau}_{\text{AIPW}}^*\\=&\frac{|\mathcal{D}^{1,1}|}{n}\hat{\tau}^{\mathcal{D}^{1,1}}+\frac{|\mathcal{D}^{1,2}|}{n}\hat{\tau}^{\mathcal{D}^{1,2}}-\frac{|\mathcal{D}^{1,1}|}{n}\hat{\tau}^{\mathcal{D}^{1,1},*}-\frac{|\mathcal{D}^{1,2}|}{n}\hat{\tau}^{\mathcal{D}^{1,2},*}\\=&\frac{|\mathcal{D}^{1,1}|}{n}\left(\hat{\tau}^{\mathcal{D}^{1,1}}-\hat{\tau}^{\mathcal{D}^{1,1},*}\right)+\frac{|\mathcal{D}^{1,2}|}{n}\left(\hat{\tau}^{\mathcal{D}^{1,2}}-\hat{\tau}^{\mathcal{D}^{1,2},*}\right)\\=&\frac{|\mathcal{D}^{1,1}|}{n}\left(\hat{m}_1^{\mathcal{D}^{1,1}}-\hat{m}_0^{\mathcal{D}^{1,1}}-\hat{m}_1^{\mathcal{D}^{1,1},*}+\hat{m}_0^{\mathcal{D}^{1,1},*}\right)\\&+\frac{|\mathcal{D}^{1,2}|}{n}\left(\hat{m}_1^{\mathcal{D}^{1,2}}-\hat{m}_0^{\mathcal{D}^{1,2}}-\hat{m}_1^{\mathcal{D}^{1,2},*}+\hat{m}_0^{\mathcal{D}^{1,2},*}\right)\\=&\frac{|\mathcal{D}^{1,1}|}{n}\left(\hat{m}_1^{\mathcal{D}^{1,1}}-\hat{m}_1^{\mathcal{D}^{1,1},*}\right)-\frac{|\mathcal{D}^{1,1}|}{n}\left(\hat{m}_0^{\mathcal{D}^{1,1}}-\hat{m}_0^{\mathcal{D}^{1,1},*}\right)\\&+\frac{|\mathcal{D}^{1,2}|}{n}\left(\hat{m}_1^{\mathcal{D}^{1,2}}-\hat{m}_1^{\mathcal{D}^{1,2},*}\right)-\frac{|\mathcal{D}^{1,2}|}{n}\left(\hat{m}_0^{\mathcal{D}^{1,2}}-\hat{m}_0^{\mathcal{D}^{1,2},*}\right)\end{aligned} \tag{23}$$

To verify $\sqrt{n}\left(\hat{\tau}_{\text{AIPW}} - \hat{\tau}_{\text{AIPW}}^*\right) \to_p 0$, it suffices to show that $\sqrt{n}\left(\hat{\tau}^{\mathcal{D}^{1,1}} - \hat{\tau}^{\mathcal{D}^{1,1},*}\right) \to_p 0$. Hence, it is further suffices to show that $\sqrt{n}\left(\hat{m}_1^{\mathcal{D}^{1,1}} - \hat{m}_1^{\mathcal{D}^{1,1},*}\right) \to_p 0$. The proof can then be extended by carrying out the same argument for different cross-fitted folds and treatments.

For the first cross-fitted fold and treatment ($w = 1$), we decompose the error term as follows:

$$
\begin{aligned}
&\hat{m}_1^{\mathcal{D}^{1,1}} - \hat{m}_1^{\mathcal{D}^{1,1},*} \\
&= \frac{1}{|\mathcal{D}^{1,1}|} \sum_{i \in \mathcal{D}^{1,1}} \left( \hat{\mu}_1^2(x_i) + a_i \frac{y_i - \hat{\mu}_1^2(x_i)}{\hat{\pi}^{1,2}(x_i)} - \mu_1(x_i) + a_i \frac{y_i - \mu_1(x_i)}{\pi(x_i)} \right) \\
&= \frac{1}{|\mathcal{D}^{1,1}|} \sum_{i \in \mathcal{D}^{1,1}} \left( \left( \hat{\mu}_1^2(x_i) - \mu_1(x_i) \right) \left( 1 - \frac{a_i}{\pi(x_i)} \right) \right) \\
&\quad + \frac{1}{|\mathcal{D}^{1,1}|} \sum_{i \in \mathcal{D}^{1,1}} a_i \left( (y_i - \mu_1(x_i)) \left( \frac{a_i}{\hat{\pi}^{1,2}(x_i)} - \frac{1}{\pi(x_i)} \right) \right) \\
&\quad - \frac{1}{|\mathcal{D}^{1,1}|} \sum_{i \in \mathcal{D}^{1,1}} \left( \left( \hat{\mu}_1^2(x_i) - \mu_1(x_i) \right) \left( \frac{1}{\hat{\pi}^{1,2}(x_i)} - \frac{1}{\pi(x_i)} \right) \right).
\end{aligned}
\tag{24}
$$

For the first term, we use the fact that, thanks to our cross-fitting construction, $\hat{\mu}_1^2$ can effectively be treated as deterministic when considering terms on $\mathcal{D}^{1,1}$. We further observe that, conditional on $\mathcal{D}^{1,2}$ and observed covariate values, these terms can be treated as the average of independent mean-zero terms. We thus yield

$$
\begin{aligned}
&\mathbb{E}\left[ \left( \frac{1}{|\mathcal{D}^{1,1}|} \sum_{i \in \mathcal{D}^{1,1}} \left( \left( \hat{\mu}_1^2(x_i) - \mu_1(x_i) \right) \left( 1 - \frac{a_i}{\pi(x_i)} \right) \right) \right)^2 \Big| \mathcal{D}^{1,2}, x_i \right] \\
&= \text{Var}\left[ \frac{1}{|\mathcal{D}^{1,1}|} \sum_{i \in \mathcal{D}^{1,1}} \left( \left( \hat{\mu}_1^2(x_i) - \mu_1(x_i) \right) \left( 1 - \frac{a_i}{\pi(x_i)} \right) \right) \Big| \mathcal{D}^{1,2}, x_i \right] \\
&= \frac{1}{|\mathcal{D}^{1,1}|^2} \sum_{i \in \mathcal{D}^{1,1}} \mathbb{E}\left[ \left( \hat{\mu}_1^2(x_i) - \mu_1(x_i) \right)^2 \left( 1 - \frac{a_i}{\pi(x_i)} \right)^2 \Big| \mathcal{D}^{1,2}, x_i \right] \\
&= \frac{1}{|\mathcal{D}^{1,1}|^2} \sum_{i \in \mathcal{D}^{1,1}} \frac{1 - \pi(x_i)}{\pi(x_i)} \left( \hat{\mu}_1^2(x_i) - \mu_1(x_i) \right)^2 \\
&\leq \frac{1}{|\mathcal{D}^{1,1}|^2} \sum_{i \in \mathcal{D}^{1,1}} \left( \hat{\mu}_1^2(x_i) - \mu_1(x_i) \right)^2 \\
&= \mathcal{O}\left( \frac{1}{n^{1+2\alpha_\mu}} \right).
\end{aligned}
\tag{25}
$$

The three equalities above follow from cross-fitting, while the two inequalities follow from the overlap assumption and consistency. The second summand in Equation 24 can also be bounded by a similar argument. Finally, for the last summand in Equation 24, we use the Cauchy-Schwarz inequality:

$$
\begin{aligned}
&\frac{1}{|\mathcal{D}^{1,1}|} \sum_{i \in \mathcal{D}^{1,1}} \left( \left( \hat{\mu}_1^2(x_i) - \mu_1(x_i) \right) \left( \frac{1}{\hat{\pi}^{1,2}(x_i)} - \frac{1}{\pi(x_i)} \right) \right) \\
&\leq \sqrt{ \frac{1}{|\mathcal{D}^{1,1}|} \sum_{i \in \mathcal{D}^{1,1}} \left( \hat{\mu}_1^2(x_i) - \mu_1(x_i) \right)^2 } \times \sqrt{ \frac{1}{|\mathcal{D}^{1,1}|} \sum_{i \in \mathcal{D}^{1,1}} \left( \frac{1}{\hat{\pi}^{1,2}(x_i)} - \frac{1}{\pi(x_i)} \right)^2 } \\
&= \mathcal{O}\left( \frac{1}{n^{\alpha_\mu + \alpha_\pi}} \right),
\end{aligned}
\tag{26}
$$

which follows from the stated converge rate. Hence, we find that this term is also $\mathcal{O}_P\left(\frac{1}{\sqrt{n}}\right)$, meaning that it is negligible in probability on the $1/\sqrt{n}$-scale as claimed. $\qquad \square$

### A.2 PROOF OF THEOREM 4.2

**Proof of Theorem 4.2.** We show that $\tau \notin C_\alpha^{\text{PP}}$ with probability of at most $\alpha$; that is,

$$\limsup_{n,N\to\infty} P\left(\mid \hat{\Delta}_\tau + \hat{\tau}_2 \mid > z_{1-\alpha/2}\sqrt{\frac{\hat{\sigma}_\Delta^2}{n} + \frac{\hat{\sigma}_{\tau_2}^2}{N}}\right) \leq \alpha. \tag{27}$$

Following Lemma 4.1, we obtain that

$$\sqrt{n}\left(\hat{\Delta}_\tau - \tau + \mathbb{E}[\tau_2]\right) \Rightarrow \mathcal{N}\left(0, \sigma_\Delta^2\right). \tag{28}$$

Then, we can apply the central limit theorem and obtain that

$$\sqrt{N}\left(\hat{\tau}_2 - \mathbb{E}[\tau_2]\right) \Rightarrow \mathcal{N}\left(0, \sigma_{\tau_2}^2\right), \tag{29}$$

where $\sigma_\Delta^2$ is the variance of $\hat{\Delta}_i = \tilde{Y}_{\hat{\eta}}(x_i) - \hat{\tau}_2(x_i)$ and $\sigma_{\tau_2}^2$ is the variance of $\hat{\tau}_2(x_i)$. Therefore, by Slutsky's theorem, we yield

$$\sqrt{N}\left(\hat{\Delta}_\tau + \hat{\tau}_2 - \mathbb{E}[\hat{\Delta}_\tau + \hat{\tau}_2]\right) = \sqrt{n}\left(\hat{\Delta}_\tau - \mathbb{E}[\hat{\Delta}_\tau]\right)\sqrt{\frac{N}{n}} + \sqrt{N}\left(\hat{\tau}_2 - \mathbb{E}[\hat{\tau}_2]\right)$$
$$\Rightarrow \mathcal{N}\left(0, \frac{1}{p}\sigma_\Delta^2 + \sigma_{\tau_2}^2\right). \tag{30}$$

This, in turn, implies

$$\limsup_{n,N\to\infty} P\left(\left|\hat{\Delta}_\tau + \hat{\tau}_2 - \mathbb{E}[\hat{\Delta}_\tau + \hat{\tau}_2]\right| > z_{1-\alpha/2}\sqrt{\frac{\hat{\sigma}}{N}}\right) \leq \alpha, \tag{31}$$

where $\hat{\sigma}$ is a consistent estimate of the variance $\frac{1}{p}\sigma_\Delta^2 + \sigma_\tau^2$. Let $\hat{\sigma} = \frac{N}{n}\sigma_\Delta^2 + \sigma_\tau^2$ which is a consistent estimate since the two terms are individually consistent estimates of the respective variances. We notice that

$$\mathbb{E}\left[\hat{\Delta}_\tau + \hat{\tau}_2\right] = \mathbb{E}\left[\sum_{i\in\mathcal{D}^1}\tilde{Y}_{\hat{\eta}}(x_i) - \sum_{i\in\mathcal{D}^1}\hat{\tau}_2(x_i) + \sum_{j\in\mathcal{D}^2}\hat{\tau}_2(x_j)\right] = \mathbb{E}\left[\sum_{i\in\mathcal{D}^1}\tilde{Y}_{\hat{\eta}}(x_i)\right] = \tau. \tag{32}$$

The last step is to combine Equation 31 and Equation 32 and then apply a union bound. We then arrive at

$$\limsup_{n,N\to\infty} P\left(\left|\hat{\Delta}_\tau + \hat{\tau}_2 - \mathbb{E}[\hat{\Delta}_\tau + \hat{\tau}_2]\right| > z_{1-\alpha/2}\sqrt{\frac{\hat{\sigma}_\Delta^2}{n} + \frac{\hat{\sigma}_{\tau_2}^2}{N}}\right) \leq \alpha. \tag{33}$$

Therefore, we have $\limsup_{n,N\to\infty} P\left(\left|\hat{\Delta}_\tau + \hat{\tau}_2\right| > z_{1-\alpha/2}\sqrt{\frac{\hat{\sigma}_\Delta^2}{n} + \frac{\hat{\sigma}_{\tau_2}^2}{N}}\right) \leq \alpha.$ □

### A.3 PROOF OF THEOREM 5.2

**Proof of Theorem 5.2.** We show that $\tau \notin C_\alpha^{\text{PP}}$ with probability at most $\alpha$; that is,

$$\limsup_{n,N\to\infty} P\left(\mid \hat{\Delta}_\tau + \hat{\tau}_2 \mid > z_{1-\alpha/2}\sqrt{\frac{\hat{\sigma}_\Delta^2}{n} + \frac{\hat{\sigma}_{\tau_2}^2}{N}}\right) \leq \alpha. \tag{34}$$

First, let use define the new auxiliary random variable $Z_i = \tilde{Y}_\pi(x_i) - \hat{\tau}_2(x_i)$. Given the fact that $\mathcal{D}^1$ is RCT dataset and known propensity score, $\tilde{Y}_\pi(x)$ and $\hat{\tau}_2(x)$ are independent functions, which means $Z_i$ are **i.i.d random variables**. Then, following the CLT, we obtain that

$$\sqrt{n}\left(\hat{\Delta}_\tau - \tau + \mathbb{E}[\tau_2]\right) \Rightarrow \mathcal{N}\left(0, \sigma_\Delta^2\right). \tag{35}$$

Then, we can apply the central limit theorem on $\mathcal{D}^2$ and obtain that

$$\sqrt{N}\left(\hat{\tau}_2 - \mathbb{E}[\tau_2]\right) \Rightarrow \mathcal{N}\left(0, \sigma_{\tau_2}^2\right), \tag{36}$$

where $\sigma_\Delta^2$ is the variance of $\hat{\Delta}_i = \tilde{Y}_\pi(x_i) - \hat{\tau}_2(x_i)$ and $\sigma_{\tau_2}^2$ is the variance of $\hat{\tau}_2(x_i)$. Therefore, by Slutsky's theorem, we yield

$$
\begin{aligned}
\sqrt{N}\left(\hat{\Delta}_\tau + \hat{\tau}_2 - \mathbb{E}[\Delta_\tau + \tau_2]\right) = & \sqrt{n}\left(\hat{\Delta}_\tau - \mathbb{E}[\Delta_\tau]\right)\sqrt{\frac{N}{n}} + \sqrt{N}\left(\hat{\tau}_2 - \mathbb{E}[\tau_2]\right) \\
\Rightarrow & \mathcal{N}\left(0, \frac{1}{p}\sigma_\Delta^2 + \sigma_{\tau_2}^2\right).
\end{aligned}
\tag{37}
$$

This, in turn, implies

$$
\limsup_{n,N\to\infty} P\left(\left|\hat{\Delta}_\tau + \hat{\tau}_2 - \mathbb{E}[\Delta_\tau + \tau_2]\right| > z_{1-\alpha/2}\sqrt{\frac{\hat{\sigma}}{N}}\right) \leq \alpha,
\tag{38}
$$

where $\hat{\sigma}$ is a consistent estimate of the variance $\frac{1}{p}\sigma_\Delta^2 + \sigma_\tau^2$. Let $\hat{\sigma} = \frac{N}{n}\sigma_\Delta^2 + \sigma_\tau^2$ which is a consistent estimate since the two terms are individually consistent estimates of the respective variances. We notice that

$$
\mathbb{E}\left[\Delta_\tau + \tau_2\right] = \mathbb{E}\left[\sum_{i\in\mathcal{D}^1}\tilde{Y}_{\pi^1}(x_i) - \sum_{i=1}^n\hat{\tau}_2(x_i) + \sum_{j=1}^N\hat{\tau}_2(x_j)\right] = \mathbb{E}\left[\sum_{i=1}^n\tilde{Y}_{\pi^1}(x_i)\right] = \tau.
\tag{39}
$$

The last step is to combine Equation 38 and Equation 39 and then apply a union bound. We then arrive at

$$
\limsup_{n,N\to\infty} P\left(\left|\hat{\Delta}_\tau + \hat{\tau}_2 - \tau\right| > z_{1-\alpha/2}\sqrt{\frac{\hat{\sigma}_\Delta^2}{n} + \frac{\hat{\sigma}_{\tau_2}^2}{N}}\right) \leq \alpha.
\tag{40}
$$

Therefore, we have $\limsup_{n,N\to\infty} P\left(\left|\hat{\Delta}_\tau + \hat{\tau}_2\right| > z_{1-\alpha/2}\sqrt{\frac{\hat{\sigma}_\Delta^2}{n} + \frac{\hat{\sigma}_{\tau_2}^2}{N}}\right) \leq \alpha.$ $\qquad\square$

# B    ADDITIONAL THEORETICAL RESULTS

In this section, we present further theoretical results regarding our method. Specifically, we show how our method can be generalized to deal with distribution shifts (see Appendix B.1), finite sample settings (see Appendix B.2), and average potential outcomes (see Appendix B.3).

## B.1    DISTRIBUTION SHIFT

In our main paper, we focus on computing prediction-powered confidence intervals when the $\mathcal{D}^1$ and $\mathcal{D}^2$ come from the same distribution. We now extend the setting to the case where $\mathcal{D}^1$ comes from $\mathbb{P}$ and the $\mathcal{D}^2$ comes from $\mathbb{Q}$, and two distributions are related by a distribution shift of covariates.

**Setting under distribution shift:** First, we assume that $\mathbb{Q}$ is characterized by *covariate shift* of $\mathbb{P}$. That is, if we denote by $\mathbb{Q} = \mathbb{Q}_X \cdot \mathbb{Q}_{A|X} \cdot \mathbb{Q}_{Y|A,X}$ and $\mathbb{P} = \mathbb{P}_X \cdot \mathbb{P}_{A|X} \cdot \mathbb{P}_{Y|X}$ the relevant marginal and conditional distributions, we assume that $\mathbb{Q}_{Y|A,X} = \mathbb{P}_{Y|A,X}$, and $\mathbb{Q}_{A|X} = \mathbb{P}_{A|X}$. As in our main paper, we calculate the measure of fit from our target population $\mathbb{Q}$ via

$$\hat{\tau} = \mathbb{E}_{\mathbb{Q}} \left[ \frac{1}{N} \sum_{j=1}^{N} \hat{\tau}_2(x_j) \right]. \tag{41}$$

The estimand from Equation 41 can be transferred to form on $\mathbb{P}$ using the Radon-Nikodym derivative. In particular, suppose that $\mathbb{Q}_X$ is dominated by $\mathbb{P}_X$ and assume that the Radon-Nikodym derivative $w(X) = \frac{\mathbb{Q}_X}{\mathbb{P}_X}(X)$ is known. Then, we can rewrite Equation 41 as

$$\tau_2^w = \mathbb{E}_{\mathbb{P}} \left[ \frac{1}{N} \sum_{j=1}^{N} \hat{\tau}_2(x_j) w(x_j) \right]. \tag{42}$$

In sum, the estimation of $\hat{\tau}$ on $\mathbb{Q}$ can be written as a reweighted function. This allows us to perform inference using the rectifier based on data sampled from $\mathbb{P}$ as before. For concreteness, we explain the estimation approach in detail. Let

$$\Delta_\tau^w = \mathbb{E}_{\mathbb{P}} \left[ \frac{1}{n} \sum_{j=1}^{n} \tilde{Y}_{\hat{\eta}}(x_i) w(x_i) - \hat{\tau}(x_i) w(x_i) \right]. \tag{43}$$

Then, the confidence interval for the above rectifier suffices for prediction-powered inference on $\tau$.

**Confidence interval covariate shift:** Let $\hat{\sigma}_{\tau_2^w}^2$ denote empirical variance of $\hat{\tau}_2^w(X)$, $\hat{\sigma}_{\Delta^w}^2$ denote empirical variance of $\hat{\Delta}_{\tau^w}$. Then, for the significance level $\alpha \in (0,1)$, the prediction-powered confidence interval is

$$\mathcal{C}_\alpha^{\mathrm{PP}} = \left( \hat{\tau}^{\mathrm{PP}} \pm z_{1-\frac{\alpha}{2}} \sqrt{\frac{\hat{\sigma}_{\Delta^w}^2}{n} + \frac{\hat{\sigma}_{\tau_2^w}^2}{N}} \right), \tag{44}$$

where

$$\hat{\tau}^{\mathrm{PP}} = \hat{\Delta}_{\tau^w} + \hat{\tau}_2^w = \frac{1}{n} \sum_{j=1}^{n} \left[ \tilde{Y}_{\hat{\eta}}(x_i) w(x_i) - \hat{\tau}(x_i) w(x_i) \right] + \frac{1}{N} \sum_{j=1}^{N} \hat{\tau}_2(x_j) w(x_j), \tag{45}$$

$$\hat{\sigma}_{\Delta^w}^2 = \frac{1}{n} \sum_{i=1}^{n} \left[ \tilde{Y}_{\hat{\eta}}(x_i) w(x_i) - \hat{\tau}_2(x_i) w(x_i) - \hat{\Delta}_{\tau^w} \right]^2, \tag{46}$$

$$\hat{\sigma}_{\tau_2^w}^2 = \frac{1}{N} \sum_{j=1}^{N} \left[ \hat{\tau}_2(x_j) w(x_j) - \hat{\tau}_2^w \right]^2. \tag{47}$$

## B.2    INFERENCE IN FINITE POPULATION

Our method developed can be directly translated to finite-population settings.

**Setting under finite-population settings:** Here, we treat $\mathcal{D}^1$ and $\mathcal{D}^2$ as fixed finite populations consisting of $n$ confounder-outcome pairs, without imposing distributional assumptions on the data points. The only assumption required to apply the latter is that $\tilde{Y}_{\hat{\eta}}(x) - \hat{\tau}(x)$ has a known bound, i.e. $[a_i, b_i]$, valid for all $i \in [n]$.

In the finite-population setting, we still follow the same way of constructing the prediction-powered estimates of ATE via

$$\hat{\tau}^{\text{PP}} = \hat{\Delta}_\tau + \hat{\tau}_2 = \frac{1}{n} \sum_{j=1}^n \left[ \tilde{Y}_{\hat{\eta}}(x_i) - \hat{\tau}(x_i) \right] + \frac{1}{N} \sum_{j=1}^N \hat{\tau}_2(x_j). \tag{48}$$

**Confidence interval finite-population settings:** Let $\hat{\sigma}_{\tau_2}^2$ denotes empirical variance of $\hat{\tau}_2(X)$, $\hat{\sigma}_\Delta^2$ denotes empirical variance of $\hat{\Delta}_\tau$. Then, for significance level $\alpha \in (0, 1)$, by Hoeffding's inequality, the prediction-powered confidence interval is

$$\mathcal{C}_\alpha^{\text{PP}} = \left( \hat{\tau}^{\text{PP}} \pm \left[ \sqrt{\frac{\sum_{i=1}^n (b_i - a_i)^2}{2n^2} \log \frac{2}{\alpha}} + z_{1-\frac{\alpha}{2}} \sqrt{\frac{\hat{\sigma}_{\tau_2}^2}{N}} \right] \right), \tag{49}$$

where $\hat{\sigma}_\Delta^2 = \frac{1}{n} \sum_{i=1}^n \left( \tilde{Y}_{\hat{\eta}}(x_i) - \hat{\tau}_2(x_i) - \hat{\Delta}_\tau \right)^2$ and $\hat{\sigma}_{\tau_2}^2 = \frac{1}{N} \sum_{j=1}^N \left( \hat{\tau}_2(x_j) - \hat{\tau}_2 \right)^2$.

### B.3 INFERENCE OF AVERAGE POTENTIAL OUTCOMES

In this section, we show how to generalize our method to the average potential outcome (APO). We define the mean outcome function in $\mathcal{D}^1$ as $f_1^a(X) := \mathbb{E}[Y(a) \mid X]$.

**APO estimation.** Let $\hat{f}_1^a(x)$ be the estimated potential outcome function on $\mathcal{D}^1$ and $\hat{f}_2^a(x)$ be the estimated potential outcome function on $\mathcal{D}^2$. Let the rectifier $\Delta_a$ denotes the difference between $\hat{f}_1^a(x)$ and $\hat{f}_2^a(x)$ on $\mathcal{D}^1$, i.e., $\Delta_a = \mathbb{E}\left[ \hat{f}_1^a(x) - \hat{f}_2^a(x) \right]$, and let $\hat{\Delta}_i = \hat{f}_1^a(x_i) - \hat{f}_2^a(x_i)$. Then, the prediction-powered estimation of the APO on $\mathcal{D}^1$ is given by as

$$\hat{\mu}_{a,1}^{\text{PP}} = \hat{\Delta} + \hat{\mu}_{a,2} = \frac{1}{n} \sum_{i=1}^n \hat{\Delta}_i + \frac{1}{N} \sum_{j=1}^N \hat{f}_2^a(x_j) = \frac{1}{n} \sum_{i=1}^n \left[ \hat{f}_1^a(x_i) - \hat{f}_2^a(x_i) \right] + \frac{1}{N} \sum_{j=1}^N \hat{f}_2^a(x_j). \tag{50}$$

**Confidence interval for APO:** Let $\hat{\sigma}_{a,2}^2$ denote empirical variance of $\hat{f}_2^a(X)$, and let $\hat{\sigma}_\Delta^2$ denote empirical variance of $\hat{\Delta}_a$. Then, for significance level $\alpha \in (0, 1)$, the prediction-powered confidence interval is

$$\mathcal{C}_\alpha^{\text{PP}} = \left( \hat{\mu}_{a,1}^{\text{PP}} \pm z_{1-\frac{\alpha}{2}} \sqrt{\frac{\hat{\sigma}_\Delta^2}{n} + \frac{\hat{\sigma}_{a,2}^2}{N}} \right), \tag{51}$$

where $\hat{\sigma}_\Delta^2 = \frac{1}{n} \sum_{i=1}^n \left( \hat{f}_1^a(x_i) - \hat{f}_2^a(x_i) - \hat{\Delta}_a \right)^2$ and $\hat{\sigma}_{\tau_2}^2 = \frac{1}{N} \sum_{j=1}^N \left( \hat{f}_2^a(x_i) - \hat{\mu}_{a,2} \right)^2$.

## C  Mathematical Background

We offer a brief overview of PPI (Angelopoulos et al., 2023a). In the following of standard PPI framework, one assumes a labeled dataset $\mathcal{S}_n = \{(X_1, Y_1), \ldots, (X_n, Y_n)\}$ of $n$ i.i.d. samples drawn from some unknown, but fixed distribution $\mathbb{P}$, where $X_i \in \mathcal{X}$ is input and $Y_i \in \mathcal{Y}$ is the outcome. One further assumes a larger sample $\tilde{\mathcal{S}}_N = \{(\tilde{X}_1, f(\tilde{X}_1)), \ldots, (\tilde{X}_N, f(\tilde{X}_N))\}$ where $n \ll N$, for which the outcome is not available, but where one has access to a pre-trained function $f : \mathcal{X} \to \mathcal{Y}$.

**PPI protocol:**[8] The objective is then to estimate a statistical quantity of interest given by the estimand $\theta^*$ (e.g., the mean). In PPI, one then constructs a prediction-powered estimate $\hat{\theta}^{\text{PP}}$ through a decompisition $\hat{\theta}^{\text{PP}} = m_\theta + \sigma_\Delta$, where $m_\theta$ is called 'measure of fit' and $\Delta_\theta$ is called 'rectifier'. Of note, $m_\theta$ is typically defined by the statistical quantity of interest (e.g., $m_\theta$ computes the sample average when $\theta^*$ is the mean), while the rectifier is a measure of the prediction accuracy of $f$. Yet, the rectifier is *not* given 'out-of-the-box' but it needs to be carefully derived for the statistical quantity of interest. Finally, the prediction-powered CI is constructed via $\mathcal{C}_\alpha^{\text{PP}} = \{\theta \mid |m_\theta + \Delta_\theta| \leq w_\theta(\alpha)\}$ where $w_\theta(\alpha)$ is a constant that depends on the confidence level. Then, $\mathcal{C}_\alpha^{\text{PP}}$ is guaranteed to contain the true parameter $\theta^*$ with probability at least $1 - \alpha\%$ (Angelopoulos et al., 2023a). Crucially, the prediction-powered CI is smaller than the classical CI when the model $f$ is sufficiently accurate.

**Example:** Let us focus on the mean, i.e., $\theta^* = \mathbb{E}[Y_i]$. The classical estimate of $\theta^*$ is the sample average of the outcomes in $\mathcal{S}_n$, i.e., $\theta^{\text{class}} = \frac{1}{n} \sum_{i=1}^n Y_i$. Then, one can derive the prediction-powered estimate of the mean via

$$\hat{\theta}^{\text{PP}} = \underbrace{\frac{1}{N} \sum_{i=1}^N f(\tilde{X}_i)}_{:=m_\theta} + \underbrace{\frac{1}{n} \sum_{i=1}^n Y_i - f(\tilde{X}_i)}_{:=\Delta_\theta}, \tag{52}$$

so that 95% confidence intervals for $\theta^*$ are given by $\mathcal{C}_{95\%}^{\text{PP}} = \left( \hat{\theta}^{\text{PP}} \pm 1.96 \sqrt{\frac{\hat{\sigma}_{f-Y}^2}{n} + \frac{\hat{\sigma}_f^2}{N}} \right)$, where $\hat{\sigma}_{f-Y}^2$ and $\hat{\sigma}_f^2$ are the estimated variances of the $f(X) - Y$ and $f(\tilde{X})$, respectively. Then, the prediction-powered CI is smaller than the classical CI, since, for $n \ll N$, the width of the prediction-powered CI is primarily determined by the term $\hat{\sigma}_{f-Y}^2$. Furthermore, when the model has only small errors, we have $\hat{\sigma}_{f-Y}^2 \ll \hat{\sigma}_Y^2$, which thus helps significantly shrink the CIs.

Of note, the rectifier must be carefully tailored for the estimand, and the derivation is typically non-trivial, especially in order to obtain further theoretical guarantees (e.g., to show that the CIs are asymptotically valid).

---

[8] For a formal derivation of the CIs, we refer to Angelopoulos et al. (2023a).

## D PSEUDOCODE

In our main paper, we presented the algorithm for computing the prediction-powered estimation of ATE and confidence interval from multiple observational datasets. Here, we provide the pseudocode in Algorithm 1. We further provide the pseudocode for the extension of our method that aims at RCT+observational data (Algorithm 2).

---

**Algorithm 1** Prediction-powered CIs for ATE estimation from multiple observational datasets

---

**Input:** small dataset $\mathcal{D}^1 = \left\{ \left( x_i, a_i^1, y_i^1 \right) \right\}_{i=1,\ldots,n}$, large dataset $\mathcal{D}^2 = \left\{ \left( x_j, a_j^2, y_j^2 \right) \right\}_{j=1,\ldots,N'}$, significance level $\alpha \in (0,1)$
1: $\hat{\tau}_2(x) \leftarrow$ estimate CATE estimator on $\mathcal{D}^2$ by sample splitting
2: $\tilde{Y}_{\hat{\eta}}(x) \leftarrow$ non-centered IF score with estimated nuisance functions on $\mathcal{D}^1$ by cross-fitting
3: $\hat{\Delta}_\tau = \frac{1}{n} \sum_{i=1}^n \left[ \tilde{Y}_{\hat{\eta}}(x_i) - \hat{\tau}_2(x_i) \right]$ for $x_i \in \mathcal{D}^1$ ▷ rectifier on $\mathcal{D}^1$
4: $\hat{\tau}_2 \leftarrow \frac{1}{N} \sum_{j=1}^N \hat{\tau}_2(x_j)$ for $x_j \in \mathcal{D}^2$ ▷ measure of fit on $\mathcal{D}^2$
5: $\hat{\tau}^{\text{PP}} \leftarrow \hat{\tau}_2 - \hat{\Delta}_\tau$ ▷ prediction-powered estimator
6: $\hat{\sigma}_{\tau_2}^2 \leftarrow \frac{1}{N} \sum_{i=1}^N \left( \hat{\tau}_2(x_i) - \hat{\tau}_2 \right)^2$ ▷ empirical variance of CATE estimation in $\mathcal{D}^2$
7: $\hat{\sigma}_\Delta^2 \leftarrow \frac{1}{n} \sum_{i=1}^n \left( \hat{\Delta}_i - \hat{\Delta}_\tau \right)^2$ ▷ empirical variance of rectifier in $\mathcal{D}^1$
8: $w_\alpha \leftarrow z_{1-\alpha/2} \sqrt{\frac{\hat{\sigma}_\Delta^2}{n} + \frac{\hat{\sigma}_{\tau_2}^2}{N}}$ ▷ normal approximation
**Output:** prediction-powered confidence interval $\mathcal{C}_\alpha^{\text{PP}} = \left( \hat{\tau}^{\text{PP}} \pm w_\alpha \right)$

---

**Algorithm 2** Prediction-powered ATE estimation with RCT + observational datasets

---

**Input:** small RCT dataset $\mathcal{D}^1 = \left\{ \left( x_i, a_i^1, y_i^1 \right) \right\}_{i=1,\ldots,n}$, large dataset $\mathcal{D}^2 = \left\{ \left( x_j, a_j^2, y_j^2 \right) \right\}_{j=1,\ldots,N'}$, significance level $\alpha \in (0,1)$
1: $\hat{\tau}_2(x) \leftarrow$ estimate CATE estimator from $\mathcal{D}^2$
2: $\tilde{Y}_{\hat{\eta}}(x) \leftarrow$ estimate non-centered influential function score from $\mathcal{D}^1$ by cross-fitting
3: $\hat{\Delta}_i \leftarrow \tilde{Y}_{\hat{\eta}}(x_i) - \hat{\tau}_2(x_i)$
4: $\hat{\tau}_2 \leftarrow \frac{1}{N} \sum_{i=1}^N \hat{\tau}_2(x_j)$, and $\hat{\Delta}_\tau \leftarrow \frac{1}{n} \sum_{i=1}^n \hat{\Delta}_i$
5: $\hat{\tau}^{\text{PP}} \leftarrow \hat{\tau}_2 - \hat{\Delta}_\tau$ ▷ prediction-powered estimator
6: $\hat{\sigma}_{\tau_2}^2 \leftarrow \frac{1}{N} \sum_{j=1}^N \left( \hat{\tau}_2(x_j) - \hat{\tau}_2 \right)^2$ ▷ empirical variance of CATE estimation in $\mathcal{D}^2$
7: $\hat{\sigma}_\Delta^2 \leftarrow \frac{1}{n} \sum_{i=1}^n \left( \hat{\Delta}_i - \hat{\Delta}_\tau \right)^2$ ▷ empirical variance of rectifier in $\mathcal{D}^1$
8: $w_\alpha \leftarrow z_{1-\frac{\alpha}{2}} \sqrt{\frac{\hat{\sigma}_\Delta^2}{n} + \frac{\hat{\sigma}_{\tau_2}^2}{N}}$ ▷ normal approximation
**Output:** prediction-powered confidence interval $\mathcal{C}_\alpha^{\text{PP}} = \left( \hat{\tau}^{\text{PP}} \pm w_\alpha \right)$

---

# E    EXTENDED LITERATURE REVIEW

In this section, we present additional related work on uncertainty quantification in causal treatment effect estimation.

**Uncertainty quantification for causal quantities:** Various approaches exist for quantifying uncertainty in causal estimates, with many relying on Bayesian methods (e.g., Alaa & Van Der Schaar, 2017; Hess et al., 2024; Jesson et al., 2020; Horii & Chikahara, 2024). While effective, Bayesian methods require prior distributions informed by domain knowledge, making them less robust to model misspecification and unsuitable for model-agnostic machine learning frameworks. Other works quantify the uncertainty in treatment effect estimation through estimating and sampling from the conditional distribution of the treatment effect (Melnychuk et al., 2024). Even other techniques offer finite-sample uncertainty guarantees through conformal prediction (e.g., Lei & Candès, 2021; Schröder et al., 2024) for potential outcomes. However, uncertainty intervals for treatment effects constructed from conformal prediction intervals around the potential outcomes commonly tend to be very wide. In contrast, in our work, we aim to provide *narrow* confidence intervals for the average treatment effect.

**CIs for ATE:** The classical way of constructing confidence intervals by making use of one observational dataset is utilizing the TMLE estimator and the AIPW estimator (Bang & Robins, 2005; van der Laan & Rubin, 2006). This idea is based on the property of unbiasedness and bounded variance, which provide the theoretical support for valid CIs. Hatt & Feuerriegel (2021) propose a novel regularization framework for estimating ATEs that exploits unconfoundedness.

**CIs for ATE from multiple datasets:** Other works aim at combining observational datasets to estimate ATEs (Yang & Ding, 2020; Guo et al., 2022). However, these works make assumptions that the small dataset needs to be sampled from the same distribution as the observational dataset. Also, these studies aim to create more efficient point estimators. They use bootstrap way to build confidence intervals, yet which leads to more uncertainty in the estimates.

Another important stream of combining multiple datasets Kallus et al. (2018) proposed a method that combined the RCT dataset and observational dataset to obtain an estimate of CATE, which could be seen as a special case of our method in Section 5. Demirel et al. (2024) also applied prediction-powered inference but focused on average potential outcomes and did not consider the uncertainty quantification of estimation as we do.

## F  EXPERIMENTAL DETAILS

### F.1  SYNTHETIC DATASET

Following the setup in Section 6.1, we consider the three different scenarios of confounding in $\mathcal{D}^2$. As shown in Equation 6.1, a larger value of $\alpha_u$ and a smaller value of $l_u$ imply stronger confounding components. For scenario 1, we set $\alpha_u = 0$ and $l_u = 10^6$, which is a scenario that is almost no confounding. For scenario 2, we set $\alpha_u = 0$ and $l_u = 0.5$, which means that we still do not consider the linear component of $U$, but let the unobserved confounder play a more important role in the exponential term. For scenario 3, we set $\alpha_u = 10$ and $l_u = 0.5$, where the unobserved confounder both influences linear and exponential terms; thus, it presents a scenario with strong confounding. For a better understanding, we state three kernels of the different settings in the following:

$$k_{\text{scenario1}}\left((X,U),(X',U')\right) = \exp\left[-\frac{(X-X')}{2 \times 10^6} - \frac{(U-U')}{2 \times 10^6}\right], \tag{53}$$

$$k_{\text{scenario2}}\left((X,U),(X',U')\right) = \exp\left[-\frac{(X-X')}{2 \times 10^6} - \frac{(U-U')}{1}\right], \tag{54}$$

$$k_{\text{scenario3}}\left((X,U),(X',U')\right) = 10 \times UU' + \exp\left[-\frac{(X-X')}{2 \times 10^6} - \frac{(U-U')}{1}\right]. \tag{55}$$

Here, we also need to make clear that the unobserved confounder only plays a role in the data-generating process of treatment, which means that does not have a direct relationship with with the difference in the means across $\mathcal{D}^2$ and $\mathcal{D}^1$.

### F.2  MEDICAL DATASET

We follow the prior research (Melnychuk et al., 2022; Frauen et al., 2023) to demonstrate our method based on the MIMIC-III dataset (Johnson et al., 2016), which includes electronic health records (EHRs) from patients admitted to intensive care units. We extract 8 confounders (heart rate, sodium, red blood cell count, glucose, hematocrit, respiratory rate, age, gender) and a binary treatment (mechanical ventilation) using an open-source preprocessing pipeline (Wang et al., 2020). We define the outcome variable as the red blood cell count after treatment. To extract features from the patient trajectories in the EHRs, we sample random time points and average the value of each variable over the ten hours prior to the sampled time point. All samples with missing values and outliers are removed from the dataset. Our final dataset contains 14719 samples, which we separate the samples now into two datasets $\mathcal{D}^1$ and $\mathcal{D}^2$ with a constant ratio $n/N = 1/50$ and add noise on $\mathcal{D}^2$.

For the second semi-synthetic dataset, we study COVID-19 hospitalizations in Brazil across different regions (Baqui et al., 2020). We are interested in predicting the effect of comorbidity on the mortality of COVID-19 patients. To model two different observational datasets, we use the regions of the hospitals in Brazil, which are split into North and Central-South. As observed confounders, we include age, sex, and ethnicity. Further, we exclude patients younger than 20 or older than 80 years. To model comorbidity as a binary variable, we define comorbidity as 1 if at least one of the following conditions were diagnosed for the patient: cardiovascular diseases, asthma, diabetes, pulmonary disease, immunosuppression, obesity, liver diseases, neurological disorders, and renal disease. We then use the same data-generating process in Section 6.1 to generate $A_i$ and $Y_i$, while using the second confounding scenario and keeping the ratio of sample size $n/N = 1/50$ and $n + N = 6881$.

### F.3  IMPLEMENTATION DETAILS

We choose the DR-learner to compute $\hat{\tau}_2$ in $\mathcal{D}^2$ together with linear regression and logistic regression models for estimating the nuisance functions. We also use linear regression and logistic regression models for the nuisance function regression when estimating the $\hat{\tau}^{\text{AIPW}}$. We use the default settings fo their regression models, and we did not perform any hyperparameter optimization, as our method aims to provide an agnostic confidence interval that applies to all CATE estimators. All our experiments are based on average results from runs across five random seeds.

# G ADDITIONAL EXPERIMENTAL RESULTS

## G.1 NEURAL INSTANTIATIONS OF OUR METHOD

We follow the same experiment setting and data generation process in Section 6.1, but replace the regression model for the nuisance regression model with a multi-layer perception (MLP) in Figure 5. Compared with the simple linear regression, our method achieves CIs that have a shorter width (as desired). Further, the CIs from our method consistently cover the oracle ATE, which again confirms the superiority of our method.

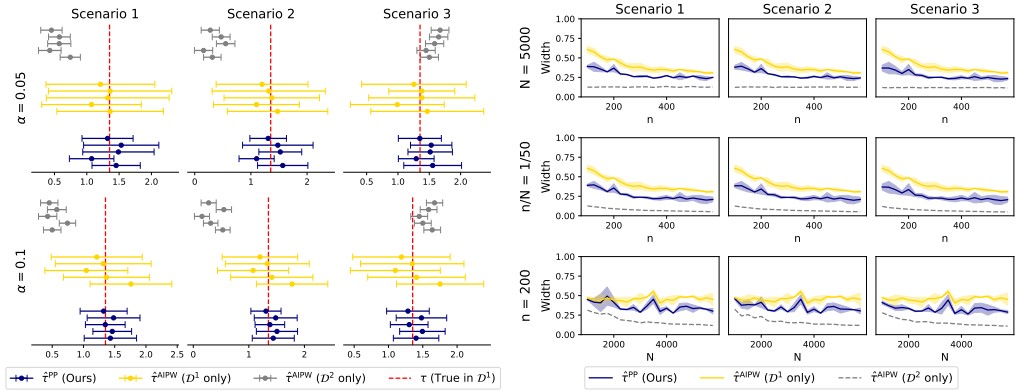

Figure 5: Results for MLP as regression method.

## G.2 Instantiation with other machine learning models

Figure 6, we follow the same data generated setting as experiments in the Figure 3 but replace the regression method used for the nuisance parameter estimation from a linear regression to XGBoost. Again, our method is highly effective, which demonstrates the flexibility of our method beyond a simple regression model.

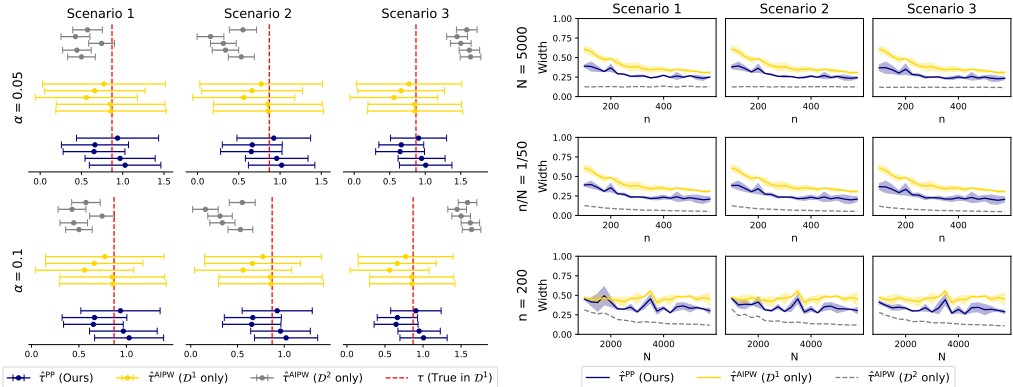

Figure 6: Results for using XGBoost as nuisance parameter regression model.

## G.3 HIGH-DIMENSIONAL COVARIATES

We repeated our experiments with more input variables to show that our method is robust in settings with high-dimensional covariate spaces. For this, we used a data-generating mechanism similar to that in the main paper. In Figure 7, we generate 5 covariates, $x \in [-1, 1]^5$. In Figure 8, we generate 50 covariates, $x \in [-1, 1]^{50}$. In Figure 9, we generate 500 covariates, $x \in [-1, 1]^{500}$. The results show that the CIs from our method consistently cover the oracle ATE and that our method reduces the width of CIs (as desired).

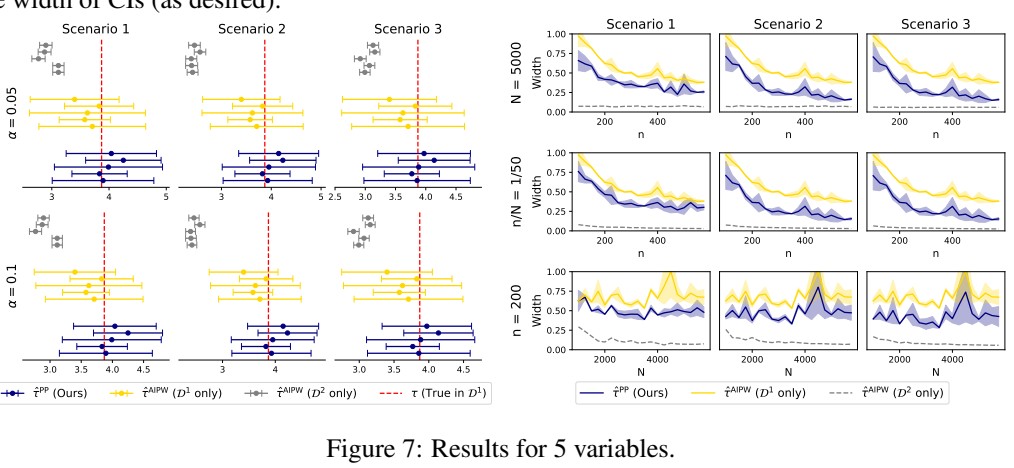

Figure 7: Results for 5 variables.

Figure 8: Results for 50 variables.

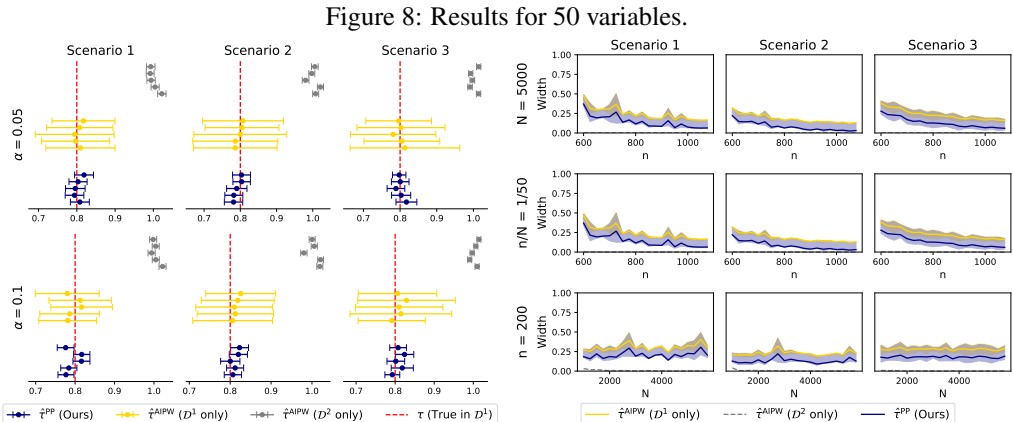

Figure 9: Results for 500 variables.

## G.4 STRENGTH OF DEPENDENCE

As extended experiments based on Section G.3, we simulated the $x \in [-1, 1]^4$ and let $x_5 = \frac{1}{n} \sum_{i=1}^{4} x_i$, which leads to collinearity in the input space in Figure 10. Thereby, we can assess the sensitivity of our method to a varying strength of dependence in the input space. Compared with i.i.d. high-dimensional covariates, we notice that the dependence does not affect our method. Our method still outperforms the other baselines and achieves the best CI width.

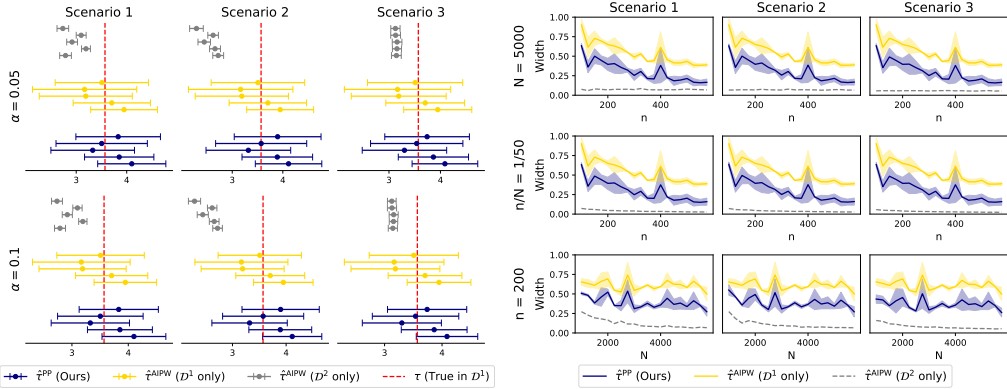

Figure 10: Results for varying dependence strength in input space.

## G.5 DIFFERENT STRENGTHS OF (UN)CONFOUNDING IN $\mathcal{D}^1$

We aim to show the experiment setting when relaxing 'unconfoundedness' assumption for $\mathcal{D}^1$. We fixed the confoundedness in $\mathcal{D}^2$ as in Scenario 2 but varied the confoundedness in $\mathcal{D}^1$ from Scenario 1 to 3 in Figure 11. We noticed that, while the strength of confounding becomes larger, our method performs better. The results again confirm that our method performs best.

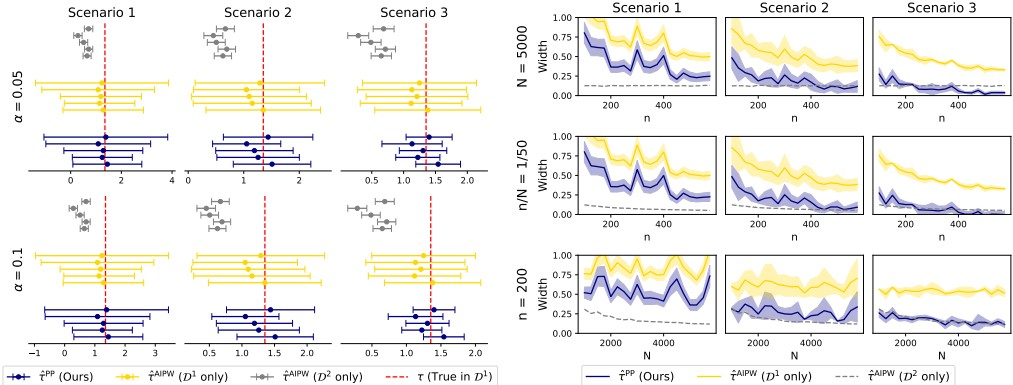

Figure 11: Results for relaxing unconfoundedness assumptions in $\mathcal{D}^1$.

### G.6 ROBUSTNESS CHECK OF APPLYING OUR AIPW METHOD TO RCT+OBSERVATIONAL DATASETS

**Data:** We adopt the same data-generating process as outlined in the main paper while applying our proposed AIPW method described in Section 4 to the RCT+observational setting.

**Main results:** In Figure 12, we demonstrates that, when replacing the known propensity score with the estimated propensity score, the performance difference is small. Both methods consistently cover the oracle ATE (in the left figure) and show a large gain compared to the naïve baseline (in the right figure).

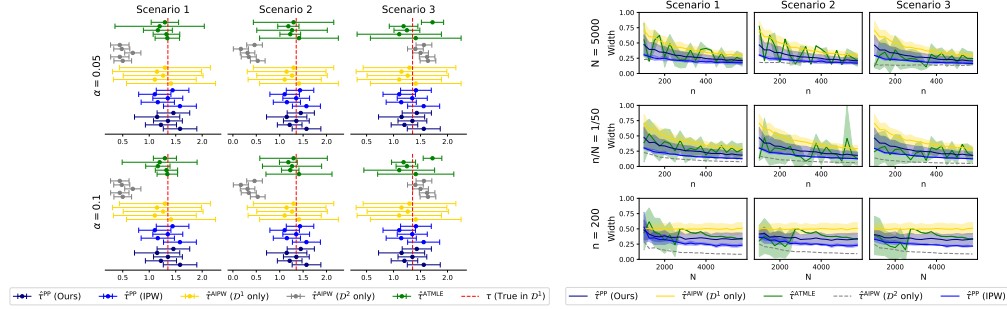

Figure 12: Applying our AIPW method to RCT+observational datasets.

### G.7 REFUTATION CHECK OF APPLYING THE A-TMLE METHOD TO OBSERVATIONAL+OBSERVATIONAL DATASETS

We apply the A-TMLE method to the synthetic datasets with observational+observational data. Of note, this violates the assumptions that underly A-TMLE, so we expect that the method leads to large errors.

In Figure 13, we notice that, when applying the A-TMLE method to the synthetic dataset, the A-TMLE performs not that well. Although it constructs short CIs, A-TMLE barely covers the oracle ATE in the left figure. In the right figure, the A-TMLE method shows a large instability in the estimation process again. These findings highlight that A-TMLE leads to CIs that are *not* faithful in RCT+observational settings. Again, this is expected.

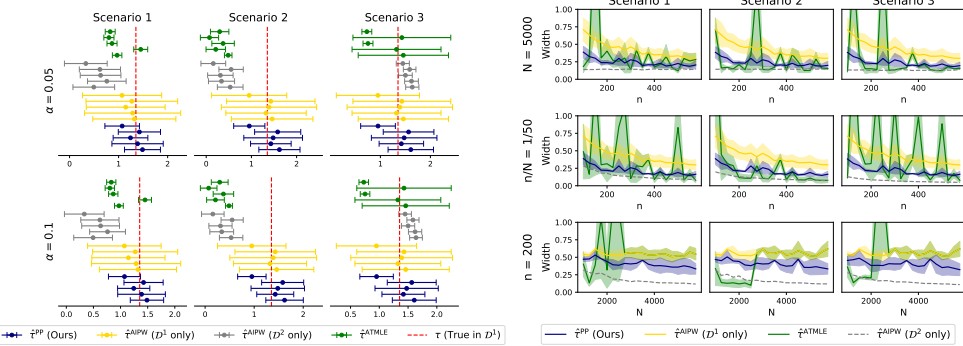

Figure 13: Applying the A-TMLE to multiple observational datasets.

## G.8 Increasing sample size in $\mathcal{D}^1$

**Data:** To provide a more comprehensive evaluation of our method, we increased the sample size in $\mathcal{D}^1$ to enable further comparisons under varying conditions. In Figure 14, the sample size in $\mathcal{D}^2$ is fixed at 5000 ($N = 5000$), while the sample size in $\mathcal{D}^1$ varies from 100 to 2500 across three distinct scenarios. This setup allows us to systematically assess the performance of our method under different data regimes.

**Main results:** Figure 14 reveals that our method consistently outperforms the naïve method across all scenarios. Notably, as the sample size in $\mathcal{D}^1$ increases, the performance gap gradually narrows, indicating diminishing returns in improvement as more data becomes available in $\mathcal{D}^1$. These results are expected and, therefore, further validate the robustness of our method.

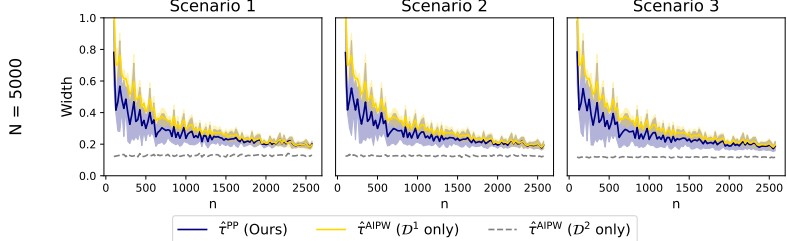

Figure 14: **Performance for an increasing sample size of $\mathcal{D}^1$.** The figure shows the width of the CIs averaged over five different seeds ($\alpha = 0.05$). Here, we vary the size of $\mathcal{D}^1$ datasets given constant sample size $N$ ($\mathcal{D}^2$) from 100 to 2500. Note that $\hat{\tau}^{\text{AIPW}}$ ($\mathcal{D}^2$ only) is shown in intentionally shown in gray: it is *not* faithful as seen in the left plot and therefore *not* a valid baseline. $\Rightarrow$ Our method continually performs better than the $\hat{\tau}^{\text{AIPW}}$ ($\mathcal{D}^2$ only).

## G.9 RMSE AND COVERAGE FOR THE EXPERIMENTS WITH SYNTHETIC DATA

In Table 2, we report the RMSE of our point estimation and the width of the CIs in Section 6.1.
Table 2: We report the RMSE of the ATE estimator and the width of the CIs. We use the synthetic dataset. The results for $\hat{\tau}^{\text{AIPW}}$ ($\mathcal{D}^2$ only) are shown in gray because the estimator is *not* faithful and therefore also *not* a viable baseline. Reported is the average performance over 5 random seeds.

| Dataset | RMSE | Width |
|---|---|---|
| $\hat{\tau}^{\text{AIPW}}$ ($\mathcal{D}^1$ only) | 0.298/0.298/0.298 | 0.241/0.240/0.237 |
| $\hat{\tau}^{\text{AIPW}}$ ($\mathcal{D}^2$ only) | 0.442/0.478/0.476 | 0.217/0.144/0.131 |
| $\hat{\tau}^{\text{PP}}$ **(Ours)** | **0.276/0.274/0.271** | **0.241/0.240/0.237** |

Smaller is better. Best value in bold.

## H COMPARISON TO A-TMLE

In this section, we compare our method against A-TMLE (van der Laan et al., 2024) and thereby highlight key differences as well as why the training in A-TMLE is unstable.

**About A-TMLE:** A-TMLE is a method that combines an RCT + observational dataset to estimate the ATE and can also construct valid confidence intervals. In van der Laan et al. (2024), the authors prove that the A-TMLE estimator is $\sqrt{n}$-consistent and asymptotically normal and gives valid confidence intervals. As a result, A-TMLE achieves smaller mean-squared errors and narrower confidence intervals.

The A-TMLE method proceeds as follows. First, in A-TMLE, the author decomposed targeted ATE estimand as the difference of (a) the pooled-ATE estimand $\tilde{\Psi}$ and (b) a bias estimand $\Psi^{\#}$, $\Psi = \tilde{\Psi} - \Psi^{\#}$. At a high level, A-TMLE constructs two separated TMLE estimators for the $\tilde{\Psi}$ and $\Psi^{\#}$. Then, A-TMLE calculates the difference of TMLE outcomes as the targeted estimand.

More specifically, for the bias estimand $\Psi^{\#}$, the estimation process can be decomposed into two steps: (i) learning a parametric working model, and (ii) constructing an efficient estimator for the targeted estimands. In the first step (i), the method applies the highly adaptive lasso minimum-loss estimator (HAL-MLE)(van der Laan et al., 2023) with the HAL basis functions for the semiparametric regression working model. Here, the method uses the 'atmle' R-package (Qiu et al., 2024). Given the above definition, one can define the working-model-specific projection parameter as

$$\Psi^{\#}(P) = E_p \Pi_p(0 \mid W, 0)\tau_{w,n,\beta(P)}(W, 0) - E_p \Pi_p(0 \mid W, 1)\tau_{w,n,\beta(P)}(W, 1), \tag{56}$$

where $P$ denotes the distribution and where $W$ denotes the covariates. We refer to van der Laan et al. (2024) for more details about the notation.

After that, we need the canonical gradient of the $\beta(P)$-component to construct the canonical gradient of the working-model-specific projection parameter $\Psi_{\mathcal{M}_{w,2}}(P)$ at $P$. However, when calculating the canonical gradient of the $\beta(P)$-component, one of the important things to observe is that $I_p = E_p \Pi(1 - \pi)(1 \mid W, A)\phi\phi^T(W, A)$, which measures the variance-covariance structure across basis functions. The expression for $I_p$ adjusts for variability in different directions, reducing weights for directions with high variance (overrepresented in data) and increasing weights where variance is low (underrepresented). Hence, A-TMLE essentially performs an adaptive weighting to make the patients in both datasets more similar for the final estimate.

**The reason for why A-TMLE is unstable:** The computation of $I_p$ has an important **shortcoming**: when (a) the dimension of the covariate space is low or (b) when collinearity among the covariates exists, the computation of $I_p$ is challenging due to the matrix inversion. Eventually, this can lead to numerical instabilities, which can cause the entire A-TMLE method to break down.

**Differences to our method:** In addition to the drawbacks of A-TMLE, two key differences exist as follows: (1) differences in ATE estimation processes and (2) differences in the flexibility of estimating $\hat{\tau}_2$. In the following, we discuss the differences (1) and (2) in detail:

(1) *Differences in the ATE estimation process.* One of the key differences between our method and A-TMLE is that our method is based on two different ATE estimations when computing the rectifier. In our method, we define the rectifier $\Delta_{\tau}$ as the difference of $\hat{\tau}_{\text{AIPW}}$ and $\hat{\tau}_2$ on $\mathcal{D}^1$. Formally, we have

$$\hat{\Delta}_{\tau} = \frac{1}{n}\sum_{i=1}^{n}\left[\tilde{Y}_{\hat{\eta}}(x_i) - \hat{\tau}_2(x_i)\right] \tag{57}$$

$$= \frac{1}{n}\sum_{i=1}^{n}\left[\left(\frac{A_i}{\hat{\pi}(x_i)} - \frac{1 - A_i}{1 - \hat{\pi}(x_i)}\right)Y_i - \frac{A_i - \hat{\pi}(x_i)}{\hat{\pi}(x_i)\left(1 - \hat{\pi}(x_i)\right)}\left[(1 - \hat{\pi}(x_i))\,\hat{\mu}_1(x_i) + \hat{\pi}(x_i)\hat{\mu}_0(x_i)\right] - \hat{\tau}_2(x_i)\right].$$

In contrast, A-TMLE defines the target estimand by applying a bias correction $\Psi^{\#}$, which can be viewed as the expectation of a weighted combination of the conditional effect of the treatment indicators on the treatment effect of the two treatment arms, where the weights are the probabilities of enrolling in the RCT of the two arms. Then, the highly adaptively lasso minimum-loss estimator (HAL-MLE) is used to learn the semi-parametric regression model.

(2) *Flexibility*. Another key difference is that our method is more flexible, allowing us to use any approach to estimate $\hat{\tau}_2$ in $\mathcal{D}^2$. In contrast, the process in A-TMLE is more rigid: A-TMLE constructs a TMLE for the pooled-ATE and bias correction term estimation. This can limit the flexibility for computing $\hat{\tau}_2$, especially when we want to use different modeling approaches for both datasets (which is likely given that one dataset is probably larger than the other!).

Instead, our method supports a variety of approaches, allowing end-users of our method to better adapt to the underlying data-generating process. For example, we can use various meta-learners like the S-learner, T-learner, R-learner, and DR-learner, where each comes with unique strengths in practice. The S-learner, for instance, works well when there are fewer treatment interactions, while the T-learner and R-learner handle more complex treatment effect patterns.

Additionally, our method allows us to use pre-trained models directly (which is unlike A-TMLE!). This allows us – in our method – to calculate the ATE from model predictions without needing to re-fit or modify the model. Alternatively, one can even use large language models or foundation models to generate the predictions of $\hat{\tau}_2$. The flexibility to use various models or integrate pre-trained models makes our approach more flexible to handle a broad variety different settings and data structures. We believe that this makes our method a powerful tool for accurate ATE estimation in a range of applications. For example, if we are given a pre-trained machine learning model $f(x)$, then we have access to the predictions on $\mathcal{D}^2$ as $\hat{f}(x)$. Formally, we then yield the measure of fit and the rectifier via

$$\hat{\tau}_2 = \frac{1}{N} \sum_{j=1}^{N} \hat{f}(x_j), \tag{58}$$

$$\hat{\Delta}_\tau = \frac{1}{n} \sum_{i=1}^{n} \left[ \tilde{Y}_{\hat{\eta}}(x_i) - \hat{f}(x_i) \right] = \frac{1}{n} \sum_{i=1}^{n} \left[ \left( \frac{A_i}{\hat{\pi}(x_i)} - \frac{1 - A_i}{1 - \hat{\pi}(x_i)} \right) Y_i \right. \tag{59}$$

$$\left. - \frac{A_i - \hat{\pi}(x_i)}{\hat{\pi}(x_i)\left(1 - \hat{\pi}(x_i)\right)} \left[ \left(1 - \hat{\pi}(x_i)\right) \hat{\mu}_1(x_i) + \hat{\pi}(x_i)\hat{\mu}_0(x_i) \right] - \hat{f}(x_j) \right],$$

$$\hat{\tau}^{\text{PP}} = \frac{1}{N} \sum_{j=1}^{N} \hat{f}(x_j) + \frac{1}{n} \sum_{i=1}^{n} \left[ \tilde{Y}_{\hat{\eta}}(x_i) - \hat{f}(x_i) \right]. \tag{60}$$

According to the central limited theorem of the predictions $f(x)$ and the asymptotical normality of the AIPW estimator, we can construct valid CI as we mentioned in the main paper. This means, we have $\mathcal{C}_\alpha^{\text{PP}} = \left( \hat{\tau}^{\text{PP}} \pm z_{1-\frac{\alpha}{2}} \sqrt{\frac{\hat{\sigma}_\Delta^2}{n} + \frac{\hat{\sigma}_{\tau_2}^2}{N}} \right)$, where $\hat{\sigma}_\Delta^2$ and $\hat{\sigma}_{\tau_2}^2$ are variance of the rectifier and measure of fit respectively.

