# OpenReview forum: "Constructing Confidence Intervals for Average Treatment Effects from Multiple Datasets"
_ICLR.cc/2025/Conference — ICLR 2025 Poster_

### Official Review · Reviewer_pijF · 2024-10-17

**Soundness:** 3
**Presentation:** 3
**Contribution:** 2
**Rating:** 6
**Confidence:** 3

**Summary:**

The authors present a method, based on prediction-powered inference, for estimating confidence intervals from two datasets, one with observed confounders and one with additionally hidden confounders. Theoretical and empirical results are provided.

**Strengths:**

- The work addresses a clear research gap with respect to existing work.
- The provided theoretical results support the method.
- The method is presented clearly, and the work is easy to follow.

**Weaknesses:**

- The current empirical analysis is limited, with mostly quantitative results and only linear models.

- Due to the required assumptions, the practical relevance of the work is not entirely clear to me.

- The conclusion is an afterthought in the current version, with no discussion of limitations or directions for future work.

**Questions:**

1. **Empirical results:**
  - Mostly qualitative results are provided for the synthetic data. I would be interested in seeing the coverage and RMSE of each method with respect to ground truth, which should be known as the data is simulated. It seems like the proposed method has a higher bias, but lower variance.
  - In my opinion, it would be interesting to experiment with settings with more data in $\mathcal{D}_1$ to see when only using $\mathcal{D}_1$ becomes better.
  - How can the RMSE be calculated for the real data, when the true ATE is unknown?
  - Unless I am mistaken, it seems that only simple learners are considered, based on linear and logistic regression (Appendix E3)? If so, it would be insightful to also compare with more advanced ML algorithms (e.g. gradient boosting).

2. **Practical significance:**
Although the work is interesting from a theoretical perspective, I wonder at its practical significance. How realistic are your assumptions? How can a practitioner verify them and trust your method?

3. **Conclusion:** A more detailed discussion of limitations and directions for future work would be appreciated.

___


**Minor points:**
- Brackets missing in the citations on lines 48 and 49.
- Gray text in Table 1 is not applied to all columns.

---

> ### Author Response · Authors · 2024-11-21
> **Response to Reviewer pijF**
>
> Thank you for your helpful review\! We took all your comments at heart and improved our paper as follows. We thus **uploaded a new PDF** where we highlighted all major changes highlighted in **blue color**.
>
> ### **Response to “Weakness”**
>
> **(W1) Limited empirical results:**
> Thanks for giving us the chance to enrich the experiment results. We give a detailed overview in our reply to question Q1 below.
>
> **Action:** We **added various new experiments** (see our **new Appendix G**).
>
> **(W2&3) Practical relevance and limitation of our work:**
> Thank you for giving us the chance to explain why our setting is standard and thus general. **First,** the assumptions are commonly referred to as the ‘standard’ assumptions valid inference of the average treatment effect \[3\].  **Second,** our assumptions are in line with prior works on making causal inferences from multiple datasets \[2, 5\].  In fact, **we make even weaker assumptions** by saying that the dataset D\_2 can have unobserved confounding. **Third,** there are many real-world applications in medicine and public policy where the assumptions are met by design. For example, the efficacy of drugs in post-approval use is typically estimated from a large observational set consisting of electronic health records (=which can have unobserved confounders and thus match our dataset D\_2). Further, any regulatory approval of drugs involves a small RCT (=which matches our assumptions for D\_1). Similarly, many interventions in public policy involve similar settings with small-scale RCT and a large-scale observational dataset. For example, such settings are common in the development sector where the effect of interventions such as having access to microloans is evaluated on people’s lives in poor countries \[1\].
>
> We give a more in-depth explanation in our response to Questions Q2 and Q3.
>
> **Action:** We explained at greater length why our assumptions are standard, why our method makes even weaker assumptions than other methods for our task, and why and where our assumptions are met in clinical practice and policy. For the latter, we also added state further concrete examples from practice.
>
> ### **Response to “Questions”**
>
> **(Q1) Empirical results:**
> Thanks for giving us the chance to enrich the experiment results. In the following, we address the three concerns of the experiments with more experimental results:
>
> * *Balance of bias and variance:* We followed your suggestion and reported the RMSE and the coverage in our synthetic datasets experiments. We see that our method performs again best.
>   This can be expected based on our theory. The reason is the following. Following our assumptions, we assume the large dataset without unconfoundedness which means that it may lead to a biased estimator. However theoretically, according to our proof in **Appendix A.2**, our proposed prediction-powered estimator provides a valid CI for the ATE in $\\mathcal{D}^1$. In short, it is an unbiased estimation , i.e. $E(\\hat{\\Delta}\_{\\tau} \+ \\hat{\\tau}\_2) \= \\tau$. This thus explains why our method is so effective.
>
> | Method  | RMSE | Coverage |
> | :---- | :---- | :---- |
> | $\\hat{\\tau}^{\\mathrm{AIPW}}$  ($\\mathcal{D}^1$  only) | 0.298/0.298/0.298 | 0.424/0.424/0.424 |
> | $\\hat{\\tau}^{\\mathrm{AIPW}}$  ($\\mathcal{D}^2$ only) | 0.442/0.478/0.476 | 0.217/0.144/0.131 |
> | "$\\hat{\\tau}^{\\mathrm{PP}}$  (Ours) | **0.276/0.274/0.271** | **0.241/0.240/0.237** |
>
> * *Increasing sample size of $\\mathcal{D}\_1$:* Thanks. We continually increase the sample size of $\\mathcal{D}\_1$ until it almost achieves similar performance. Based on your suggestion, we improved the experiments and increased the sample size further to larger values (see our **new** **Appendix G.8**). The results are as expected and confirm our theory.
> * *Comparison against the ground-truth ATE:* Thank you for pointing this out. For the real-world application, in our study, we utilize a semi-synthetic dataset, where we have knowledge of how the potential outcomes are generated. This allows us to directly evaluate the accuracy and validity of our method **by comparing the estimated ATE against the ground truth**. By leveraging the semi-synthetic nature of the dataset, we can strike a balance between real-world complexity and the ability to validate our results rigorously. Here, we see that our method is very accurate in learning the ground truth ATE.
> * *Generalization to machine learning models:* We **conducted new experiments with  XGBoost** (see our **new** **Appendix G.2**) and **with neural networks** (see **our new Appendix G.1**). Again, our method is highly effective (and the performance gain over the baseline becomes even larger\!).
>
>   **Action:** We added further clarifications for the points above, and we further **added the new experiments to our revised PDF**.

---

> ### Author Response · Authors · 2024-11-21
>
> **(Q2) Practical significance:**
> Thanks so much for your question about the practical application of our work. Estimating the confidence intervals for ATEs is a relevant question in many fields such as medicine. We also would like to emphasize that **our assumptions are realistic in practice**. In particular, we followed the standard assumptions for estimating the ATE from observational datasets in existing literature, such as Imbens 2004 \[4\]; Rubin 2006 \[6\]; Shalit et al., 2017 \[7\]. **Our work is thus consistent.**
>
> There are also papers that consider multiple datasets (see the overview in Table 1 of our revised paper). However, they always assume the combination of RCT and observational datasets \[2, 5\]. Hence, these methods make stronger assumptions than ours. In contrast, our method makes weaker assumptions. Our method proposed constructing valid confidence intervals with more released assumptions, i.e., multiple observational datasets, where we relax the assumption of unobserved confounding.
>
> **Action:** We explained at greater length why our assumptions are standard, why our method makes even weaker assumptions than other methods for our task, and why and where our assumptions are met in clinical practice and policy. For the latter, we also added state further concrete examples from practice.
>
> **(Q3) Discussion about limitations and direction of future work:**
> Thank you for the suggestion. We **added a discussion of limitations and directions for future work to our conclusion section**. For example, one avenue for improvement is to extend our method to other estimands such as CATE. Additionally, replacing the $\\tau\_2$ model with a pre-trained large language model (LLM) could offer new applications for our method and thus make it more flexible. Similarly, one could also tailor our method to representations with text to make inferences from natural language more effective. Nevertheless, as with any other paper in causal inference, certain but standard assumptions should be fulfilled, so we encourage careful and responsible use in practice.
>
> **Action:** We added a discussion of limitations and directions for future work to our conclusion section.
>
> **(Q4) Minor points:**
> Thank you for pointing out the typos.
>
> **Action:** We have corrected all the typos.
>
> **References:**
> \[1\] Abhijit Banerjee, Esther Duflo, Rachel Glennerster, and Cynthia Kinnan. The Miracle of Microfinance? Evidence from a Randomized Evaluation. ISSN 1945-7782.
> \[2\] Ilker Demirel, Ahmed Alaa, Anthony Philippakis, and David Sontag. Prediction-powered Generalization of Causal Inferences, 6 2024\.
> \[3\] Phillip Heiler and Ekaterina Kazak. Valid inference for treatment effect parameters under irregular identification and many extreme propensity scores. 222(2):1083–1108. ISSN 0304-4076.
> \[4\] Guido W. Imbens. Nonparametric estimation of average treatment effects under exogeneity: A review. Review of Economics and Statistics, 86(1):4–29, 2004\.
> \[5\] Nathan Kallus, Aahlad Manas Puli, and Uri Shalit. Removing Hidden Confounding by Experimental Grounding. Advances in neural information processing systems, 2018\.
> \[6\] Donald B. Rubin. Matched sampling for causal effects. Cambridge University Press, 2006\.
> \[7\] Uri Shalit, Fredrik D Johansson, and David Sontag. Estimating individual treatment effect: generalization bounds and algorithms. International conference on machine learning, 2017\.

---

> > ### Comment · Reviewer_pijF · 2024-11-24
> >
> > Thank you for your detailed response and helpful changes! I have updated my score.

---

> > > ### Author Response · Authors · 2024-11-25
> > >
> > > Thank you for your positive feedback! We will incorporate all action points into our revised version of paper.

---

### Official Review · Reviewer_sim7 · 2024-10-17

**Soundness:** 3
**Presentation:** 2
**Contribution:** 2
**Rating:** 6
**Confidence:** 5

**Summary:**

The authors propose a new method to compute confidence intervals of the Average Treatment Effect (ATE) of the Risk Difference based on two observational data sets, the first one of small size in which all confounders are observed, the second one being larger with potential unobserved confounders. They use the largest data set to debias the estimation of the ATE computed on the first data set. The proposed confidence interval is shown to have the correct asymptotic coverage. Experiments on a toy data set and on real-world data sets show that the proposed method leads to smaller unbiased confidence intervals.

**Strengths:**

The paper focuses on a very important topic, which is uncertainty quantification of point estimate for the ATE, based on observational data sets. Obtaining accurate confidence intervals for the ATE helps to understand the effect of a treatment (e.g., drugs). Observational data are becoming more and more common in causal inference. The main contribution of the paper is to propose a new way of computing confidence intervals based on two observational data sets. The formula is rather simple and easy to use, while it can be applied to a wide variety of estimators (in the paper, AIPW or IPW are used depending on the context).
Experiments tend to show that the proposed method is better in terms of coverage and IC length.

**Weaknesses:**

There is a potential practical interest for the confidence interval and the point estimate developed in this paper. I have two majors concerns with the paper:
- the experimental results on the toy data set are inconclusive : the input variable is of dimension one, which does not correspond to a real-world problem. For the conclusion to hold, I would expect more experiments on more complex settings (more input variables, with or without dependence, how relaxing the assumption of all observed confounders in the first data set may impact the estimation performances). Experimenting on one single input variable does not allow us to obtain a clear comprehension of the performance of the proposed method.
- Theorem 5.2 is the main theoretical result of the paper. It is claimed below that it does not directly result from the Prediction-powered Inference (PPI) framework. However, going in detail through the proof, I could not find an argument specific to the causal inference framework or the AIPW estimator. The proof relies on Central Limit Theorems of two independent quantities.
Thus, I find the novelty of the contribution mild from a theoretical perspective. Besides, in my opinion, more experiments should be carried out to assess the performance of the method.

**Questions:**

- Position of the work: it is assumed that the small data set contains all confounders. The major difference with a RCT is that the propensity score is unknown and depends on the covariate (as discussed in page 4, last paragraph). This justifies the use of IPW with the true propensity score (step B, page 7). However, numerous work have shown that even in a RCT scenario where the true propensity score is known, estimating it can reduce the variance of the ATE estimate (see, e.g., references in https://arxiv.org/pdf/2303.17102). Thus, in a RCT, one would likely use an estimation of the propensity score inside the IPW estimate. This point should be discussed and may lead to qualify statements about differences between RCT and the small data set you consider. In particular, could we apply methods developed for RCT (van der Laan et al., 2024) to the first experimental setting? Paragraph l.119-121 could be modified to make explicit the differences (if any) between RCT+observational and the considered setting observational+observational.
- Regarding the work of van der Laan et al., 2024, it is mentioned that their method is unstable due to a matrix inversion (l.509-510). I went quickly through their paper and did not find such an operation. Could you be more specific, and maybe describe with more details how this method works, as it is the main competitor?
- l.151-152 ``PP CI s smaller than the classical CI when the model $f$ is sufficiently accurate.'' Is there a specific reference for this fact?
- l.203-204: It is usually required that $0< \pi(x) < 1$.
- l.269: "The estimation needs to be both valid and unbiased to later yield valid CIs.'' What does ``valid'' mean in these contexts?
- Lemma 5.1 Assumptions are not complete/clear: there are constraints on $\alpha_{\mu}$ and $\alpha_{\pi}$, the assumption on $e(\cdot)$ is about $1/\hat{e}(\cdot)$ and not $e$ and the proof of Wager holds for crossfitted estimators. The asymptotic variance involves theoretical quantities and not their estimators. Please correct the statement. Some remarks also hold for Theorem 5.2.
- l.298  $\tilde{Y}_{\hat{\eta}}$ is built using $D_1$ and then its empirical mean (subtracted by $\hat{\tau}_2$) is computed over
$D_1$.

In the proof, it is argued that $\hat{\Delta}_{\tau}$ verifies a Central Limit Theorem which is not clear here, as the term inside the sum depends on the whole sample $\mathcal{D}_1$.

I believe the proof holds for a fixed function $\tilde{Y}_{\eta}$ but not for the estimated function. Or it needs at least clarifications.
- Equation 5, how do this new confidence interval compares to a classical one (using only $\mathcal{D}_1$). Can you show that it is stricly smaller in some specific settings (and describe them)?
- Lemma 6.1 A factor $1/n$ is missing in the definition of $\hat{\tau}_1$, $\hat{\sigma}_1^2$ cannot depend on the sample size $n$. Its expression can be made explicit as a function of the nuisance components. In the proof, $\hat{\pi}$ should be replaced by $\pi$.
- l488 In the real-world application, how the RMSE is computed, since we do not have access to the true ATE value? Please define ``factual outcome''.
- l.499 ``Takeaway: Our PPI-based method is effective for medical data.'' Please qualify this statement, as you evaluate your method on two datasets only.

Typos:
- open parenthesis l.124
- l.162 : a for is missing
- l.177 and l.334 : $p$ is used for the number of input variables and the ratio between the data set sizes.
- l.363 ``This steps computes analogous the above''
- l.409 ``We now evaluate our the effectiveness''
- l469 ``Hence, our method performs thus best.''

---

> ### Author Response · Authors · 2024-11-21
> **Response to Reviewer sim7**
>
> Thank you for your careful review and helpful comments. As you can see below, we have carefully revised our paper along with your suggestions, added various new experiments, and added a comparison with the work of van der Laan et al., 2024 with detailed theoretical proof, and conducted more experiments. We thus **uploaded a new PDF** where we highlighted all major changes highlighted in **blue color**.
>
> ### **Response to “Weakness”**
>
> **(W1) More experiments:**
> Thanks for giving us the chance to expand our experiment results. In the following, we **added new experiments** along your suggestions:
>
> 1. *More input variables:* We **expanded our experiments** and now report results with more input variables. For this, we used a data-generating mechanism similar to that in the main paper but where we now generate multiple covariates (see our **new** **Appendix G.3**). The **results confirm our existing conclusions** as well as our theoretical contributions. In particular, the results show that our method is effective even for settings with multiple input variables.
> 2. *With or without dependence:* Here, we **added** the experiments while having 4 independent covariates and the 5-th covariate is the mean of the above 4 covariates and use the 5-th covariate as a component of generating the potential outcomes (see our **new** **Appendix G.4**). Again, **our method performs best**.
> 3. *Relaxing the “Unconfoundedness” assumption in $\\mathcal{D}\_1$:* Finally, we also conducted the experiments while keeping the $\\mathcal{D}\_2$ as the medium confounded scenario with three different confounding scenarios in $\\mathcal{D}\_1$ (see our **new** **Appendix G.5**). The results again confirm our theoretical contributions and that **our method performs best**.
>
> **Action:** We **added several new experiments**: (i) with high-dimensional covariates, (ii) with and without dependence, and (iii) different confounding scenarios in $\\mathcal{D}\_1$ (see our **new Appendix G**).
>
> **(W2) Specific theoretical support:**
> Our main contribution is not the derivation of a new meta-learner but **proposed a novel way to construct valid and more accurate CIs** based on the good asymptotic property of the AIPW estimator in multiple observational datasets and IPW in RCT and observational datasets. We **still strongly believe that our paper fills an important gap in the literature and provides a method that is novel and highly relevant for practice (e.g., in medicine)**. As of now, existing methods for causal inference from multiple datasets have primarily relied on point estimates, while we shift the focus to uncertainty quantification. We also believe that our application of PPI in the context of causal inference from multiple datasets is new, thus changing the underlying way how causal inference from multiple datasets is made and thus will spur new avenues for follow-up research.
>
> **Action:** We carefully checked our paper and we spelled out our main novelty clearly: we present a novel way using AIPW estimators to construct CIs from multiple datasets with new theoretical guarantees (see **our Theorem 5.2 and Theorem 6.2**).

---

> ### Author Response · Authors · 2024-11-21
>
> ### **Response to “Questions”:**
>
> **(Q1) Position of the work:**
> Thank you. We followed your suggestion closely and performed new experiments as follows:  First, we applied our AIPW-based methods to combinations of  RCT and observational datasets (see our **new Appendix G.6**). Second, we applied the A-TMLE method on multiple observational datasets (see our **new Appendix G.7**). We can see that both experiment results show that when replacing the known propensity score with the estimated one, our method still shows the faithful and even performs better in shrinking the width of CIs. Further, as expected, A-TMLE does not perform well in the new experiments because it does not properly estimate the propensity scores.
>
> **Action:** We performed new experiments where we estimated the propensity score as you suggested. Here, we cited the references by Su et al. \[6\] and Lars van der Laan et al., 2024 \[7\] as motivation for conducting such experiments. Specifically, we applied our  AIPW-based methods to combinations of RCT and observational datasets (see our **new Appendix G.6**) and the A-TMLE method to settings with multiple observational datasets (see our **new Appendix G.7**). The results again confirm the effectiveness of our methods.
>
> **(Q2) Comparison with the work of van der Laan et al., 2024 \[7\]:**
> Thank you. We followed your suggestion and added a detailed, technical comparison to highlight key differences (see our **new Appendix H**).
> (1) Our method and van der Laan rely on **different ATE estimation processes**, where A-TMLE uses separate TMLEs to estimate pooled ATE and bias correction and our method relies on AIPW to estimate the ATEs.
> (2) The validity of CIs from A-TMLE only holds when applying TMLE as the estimation process for the pooled ATE, however our method shows higher flexibility. The validity of our proposed CIs holds for the arbitrary estimation process of $\\hat{\\tau}\_2$.
> (3) In A-TMLE, it relies on the HAL-MLE to estimate the semi-parametric regression working model. While the data are simple, it is always hard to solve the covariance matrix which leads to the failure of the whole algorithm.
>
> **Action:** We added a detailed comparison with the A-TMLE from van der Laan in our **new** **Appendix H**. Therein, we highlight key differences at a technical level.
>
> **(Q3) Reference for l. 151-152:**
> Thank you. Upon reading your comment, we realized that we should be more careful in explaining the intuition of our method. We expanded our explanations as follows. As the model $f$ is sufficiently accurate, we have the rectifier almost equal to zero, $\\hat{\\Delta} \\approx 0$. Then the variance of the rectifier is significantly smaller than the variance of estimated non-centered IF scores, $\\hat{\\sigma}\_{\\Delta}^2 \\leq \\hat{\\sigma}_{f}^2$. Given the large size of $\\mathcal{D}\_2$, the variance of the estimated conditional treatment effect can be almost overlooked, since the estimated variance should be divided by the sample size of $\\mathcal{D}\_2$. The rest follows from reference \[1\].
>
> **Action:** We **added the above explanation to our paper**. Specifically, we added a summary to revise the “Intuition behind our method” in the Introduction section of the revised paper making it easier to understand. Further, we added a new paragraph “Why is our method better than using the unconfounded dataset only?” to our **revised Section 5**.
>
> **(Q4) Requirement of overlapping assumptions:**
> Thank you, we revised how we formalize the overlap assumption.
>
> **Action:** We fixed this.
>
> **(Q5) Meaning of “valid”:**
> Thank you. We refer to Barnard et al. \[2\] and refer to a confidence interval as “valid” when the interval achieves its stated coverage probability. For example, a 95% confidence interval is valid if, under repeated sampling, it contains the true parameter value approximately $95\\%$ of the time. Validity ensures the interval accurately reflects the level of uncertainty about the estimate.
>
> **Action:**  We added a formal explanation of when we regard a CI as “valid”.
>
> **(Q6) Clarification for Lemma 5.1:**
> Thank you for spotting that the notation may have been unclear here. We later need the cross-fitted estimator of the nuisance parameters and, while their convergence rate added together is quick enough, the asymptotic normality holds for the AIPW estimator
>
> **Action**: We fixed the notation. We changed Lemma 5.1 to Remark 5.1 and wrote that it follows immediately from Wager’s book.
>
> **(Q7) Confusion about l.298:**
> Sorry for the unclear here, we use cross-fitted nuisance parameters on $\\mathcal{D}\_1$ to calculate the non-centered IF scores on $\\mathcal{D}\_1$ and then calculate the rectifier as the mean of differences.
>
> **Action:** We improved our presentation.

---

> ### Author Response · Authors · 2024-11-21
>
> **(Q8) CLM:**
> Thanks for your valuable question here and you are correct. Our method heavily depends on the asymptotic normality of the estimated rectifier in our proof of Theorem 5.2. Following the CLM, the rectifier asymptotically converges to the expectation, i.e.,
> $\\sqrt{n}\\left(\\hat{\\Delta}\_{\\tau} \- \mathbb{E}\[\\hat{\\Delta}\_\\tau\]\\right) \\Rightarrow \mathcal{N} \\left(0, \\sigma^2\_{\\Delta}\\right)$ where $\\sigma^2\_{\\Delta}$ is the variance of $\\hat{\\Delta}\_i \= \\tilde{Y}\_{\\hat{\\eta}}(X\_i) \- \\hat{\\tau}\_2(X\_i)$. As for your further worry that the rectifier is constructed only on $\mathcal{D}^1$, since we assume that $\mathcal{D}^1$ and $\mathcal{D}^2$ are sampled from the same population, so the expectation of $\\hat{\\tau}\_2(X\_i)$ are same in $\mathcal{D}^1$ and $\mathcal{D}^2$. Then by Slutsky’s theorem, we could derive the asymptotical normality of our proposed ATE estimator.
>
> **Action:** We added a new section to explain the asymptotic properties of $\tau\_2$ (see **our new Appendix A.3**).
>
> **(Q9) Proof for estimated function:**
> Thanks for your question. We follow the proof of the double robustness in Wager 2024 \[8\], where both the estimated nuisance function and the non-centered IF scores also fulfill asymptotic normality property, i.e., $\\sqrt{n}(\\hat{\\tau}\_{\\mathrm{AIPW}} \- \\tau\_{\\mathrm{AIPW}}) \\rightarrow\_{p} 0$. Then, since the non-centered IF scores are an unbiased estimation of the oracle ATE, the proof is not affected.
>
> **Action:** We added a footnote about why this is not problematic.
>
> **(Q10) Proposed new CI compared to a classical one:**
> We kindly refer to our response to (Q3).
>
> **(Q11) $1/n$ missing:**
>  Thank you.
>
> **Action:** We fixed this in the revised PDF.
>
> **(Q12) How the RMSE is computed:**
> Thank you for pointing this out. For the real-world application in our study, we utilize a semi-synthetic dataset, which means we know the potential outcome generation process. This allows us to directly evaluate the accuracy and validity of our method by comparing the estimated ATE against the ground truth. By leveraging the semi-synthetic nature of the dataset, we can strike a balance between real-world complexity and the ability to validate our results rigorously.
>
> **Action:** We clarified this in our paper.
>
> **(Q13) Qualification of our “Takeaway”:**
> Thanks for your comments. We conducted comprehensive experiments not only on two distinct datasets but also using five different random seeds to ensure robustness. Additionally, we explored various experimental sample size settings to evaluate the consistency and scalability of our method in the revised paper across different scenarios. We hope that, together with the new experiments, the takeaways are warranted.
>
> **Action:** We added various new experiments to qualify our claims in the “Takeaways”.
>
> **(Q15) Typos:**
> We sincerely apologize for the typos and greatly appreciate your careful reading. We will address these errors and correct them in the revised version of our paper.
>
> **Action:** We have corrected all the typos.
>
> **References:**
> \[1\] Anastasios N. Angelopoulos, Stephen Bates, Clara Fannjiang, Michael I. Jordan, and Tijana Zrnic. Prediction-powered inference. Science, 382(6671):669–674, 11 2023\. ISSN 0036-8075, 1095-9203.
> \[2\] G. A. Barnard. Statistical Inference, 1949\. ISSN 0035-9246.
> \[3\] Weixin Cai and Mark J. van der Laan. One-step targeted maximum likelihood for time-to-event outcomes, 2019\.
> \[4\] Ilker Demirel, Ahmed Alaa, Anthony Philippakis, and David Sontag. Prediction-powered Generalization of Causal Inferences, 6 2024\.
> \[5\] Nathan Kallus, Aahlad Manas Puli, and Uri Shalit. Removing Hidden Confounding by Experimental Grounding. Advances in neural information processing systems, 2018\.
> \[6\] Fangzhou Su, Wenlong Mou, Peng Ding, and Martin J. Wainwright. When is the estimated propensity score better? high-dimensional analysis and bias correction, 2023\.
> \[7\] Mark van der Laan, Sky Qiu, and Lars van der Laan. Adaptive-TMLE for the Average Treatment Effect based on Randomized Controlled Trial Augmented with Real-World Data, 5 2024\.
> \[8\] Stefan Wager. Causal Inference: A Statistical Learning Approach, 2024\.

---

> > ### Comment · Reviewer_sim7 · 2024-11-21
> >
> > Thank you for answering most of my comments. I still have some doubts about the following points:
> > - at the bottom of page 6, you mention that $\hat{\tau}_2$ satisfies a CLT. But $\hat{\tau}_2$ is estimated and evaluated on the same data set. A CLT follows directly if $\hat{\tau}_2$ is considered as fixed, which is not the case here. This deserves more explanations. Besides, $\tau_1$ should be $\hat{\tau}_1$ and the definition of this quantity is not introduced before the bottom of page 6 (it is introduced in the sequel).
> > - In Theorem 4.2, the rate of convergence should be on $1/\hat{\pi}(x)$ and not on $\pi(x)$, the type of convergence (in probability) must be stated. Besides, if I am not mistaken, the proof of Wager (2024) holds for cross-fitted estimators, which is mentioned at all in Theorem 4.2.
> > - I do not worry about the bias of $$\hat{\Delta}_{\tau}$$
> >
> > but I wonder how you can apply a CLT to
> >
> > $\hat{\Delta}_{\tau} = \frac{1}{n} \sum_{i=1}^n (\tilde{Y}_{\hat{\eta}}(x_i) - \hat{\tau}_2(x_i))$.
> >
> > Remark 4.1 support the fact that a CLT can be applied to the first part, while a CLT holds for the second part, as the evaluation of $\hat{\tau}_2$ is performed on $D_1$. But it remains to show that the two quantities are independent to obtain (i) an asymptotic Gaussian distribution (ii) with the correct variance. Did I miss something?
> >
> > Besides, thank you for the additional simulations. I still think that $5$ variables is not sufficient to highlight the benefits of the method. I definitely would not say that this is a "high-dimensional covariate space" as stated in appendix G.3 or in your answer. Nevertheless, I appreciate your work on this rebuttal.

---

> > > ### Author Response · Authors · 2024-11-22
> > >
> > > Thank you for the fast and helpful response to our rebuttal and for enabling a constructive discussion. We are glad that our rebuttal addressed some of your concerns, and **we are confident that we can address the remaining ones below**. Again, **we have updated our PDF** and highlighted the new materials in **red color** (to show our additions from the previous edits that were in blue color).
> > >
> > > ### **Response to “Theory”:**
> > >
> > > 1. Thanks for your valuable question here, and you are correct. We forgot to add the important details that we **performed sample splitting** and thus split **$\\mathcal{D\_2}$ into two independent datasets**, one for training the fixed function $\\hat{\\tau}_2(\\cdot)$, and the other one for evaluating $\\hat{\\tau}_2(\\cdot)$ and getting the CATE estimations. Of note, we do it in our experiments rightly. **In our experiments, we split $\\mathcal{D\_2}$ into two independent datasets**, denoted as  $\\mathcal{D\_2^1}$ and $\\mathcal{D\_2^2}$. We first trained our CATE estimator (e.g., DR-learner), in $\\mathcal{D\_2^1}$ as a fixed function $\\hat{\\tau}_2(\\cdot)$. Next, we apply $\\hat{\\tau}_2(\\cdot)$ to the $\\mathcal{D\_2^2}$ and computed the average over $\\mathcal{D\_2^2}$ as $\\hat{\\tau}\_2 \= \\frac{1}{N}\\sum\_{i=1}^N\\hat{\\tau}\_2(x\_i)$. Since the $\\hat{\\tau}_2(\\cdot)$ is trained and evaluated on two **independent datasets**, $\\hat{\\tau}\_2$ satisfies a CLT. Sorry for the imprecision in Our paper.
> > >
> > > **Action:** We corrected this in our revised paper and state that we used sample splitting to train and evaluating the $\\hat{\\tau}\_2$.
> > >
> > > 2. We appreciate your suggestions. We **corrected the notation and emphasized our use of cross-fitting** in Theorem 4.2. Then, following Wager’s book, $\\hat{\\tau}^{\\mathrm{AIPW}$ is asymptotically normally distributed to the oracle ATE with corrected variance.
> > >
> > >
> > > **Action:** We revised this in our paper.
> > >
> > > 3. Regarding the asymptotic normality of $\\hat{\\Delta}_{\\tau}$,  it holds because $\\mathcal{D\_1}$ and $\\mathcal{D}\_2^1$ are two **independent datasets**.
> > >
> > >    Follow the $\\hat{\\Delta}_{\\tau} \= \\frac{1}{n} \\sum{i=1}^n (\\tilde{Y}\_{\\hat{\\eta}}(x\_i) \- \\hat{\\tau}\_2(x\_i))$, where $\\tilde{Y}\_{\\hat{\\eta}}(x\_i)$ are non-centered influence function score we estimated with cross-fitting AIPW estimation and $\\hat{\\tau}\_2(x\_i))$ are estimated on $\\mathcal{D}\_2^1$ with trained fixed function $\\hat{\\tau}_2(\\cdot)$ on $\\mathcal{D}\_2^1$.
> > >
> > >    Furthermore, as the nuisance functions of pseudo-outcomes and $\\hat{\\tau}\_2$ are trained on independent datasets, these two components are indeed independent. Since the CLT applies to the second term of $\\hat{\\Delta}\_{\\tau}$ and $\\hat{Y}\_{\\tilde{\\eta}}(x)$ and $\\hat{\\Delta}\_{\\tau}$ are independent, then the asymptotical normality holds for $\\hat{\\Delta}\_{\\tau}$.
> > >
> > >
> > > **Action:** We **fixed the notation** in Theorem 4.2 and clarified that we need the **dataset splitting** for training and evaluating of $\\hat{\\tau}\_2$.
> > >
> > > ### **Response to “Experiments”:**
> > >
> > > Thank you. We followed your suggestion and **expanded our experiments** further. We now report results with 50 and 500 input variables (see our **new** **Appendix G.3**). The **results confirm our existing conclusions** and theoretical contributions. In particular, the results show that our method is effective even for settings with higher dimensional input variables.
> > >
> > > **Action:** We followed your suggestion, and we **added experiments to show our effectiveness** for high-dimensional input variables.

---

> ### Comment · Reviewer_sim7 · 2024-11-25
>
> Thank you for your response! I only have one comment left:
> - Regarding the point 3) above, I am still not sure how you properly obtain a CLT. In order to combine the two CLT, you need to have that
>
> $$ \frac{1}{n} \sum_{i=1}^n \tilde{Y}_{\hat{\eta}}(x_i) $$
>
> and
>
> $$ \frac{1}{n} \sum_{i=1}^n \hat{\tau}_2(x_i) $$
>
> are independent, which is not the case as they both depend on $x_i$. However, I agree that the two functions $\tilde{Y}_{\hat{\eta}}(\cdot)$ and
>
> $\hat{\tau}_2(\cdot)$
>
> are independent. Can you justify more precisely how you obtain the final CLT for $\hat{\Delta}_{\tau}$?
>
>
> (on a minor note, I believe that the use of cross-fitting should be explicitly mentioned in Theorem4.2. In the current version, it is said 'sample splitting in splitted data sets' which is not very clear).

---

> ### Author Response · Authors · 2024-11-26
>
> **Thank you for updating the score! And thanks again for your valuable question and the active discussion!** We are happy to provide additional justification for the application of the  CLT for $\hat{\Delta}{\tau}$ step by step as follows:
>
> 1. To see why the CLT holds, let us define **a new auxiliary random variable** $Z_i = \tilde{Y}_{\hat{\eta}}(x_i) - \hat{\tau}_2(x_i)$. In particular, **we do not apply the CLT separately** on both summands but on the **joint** variable $Z_i$.
>
> 2. The $Z_i$ are **i.i.d. random variables** because we used cross-fitting and estimated the nuisance functions $\hat{\eta}$ on $\mathcal{D}^1$ and the CATE estimator $\hat{\tau}_2$ on $\mathcal{D}^2$ (in particular, not on $\mathcal{D}^1$). **Hence, the CLT holds for $Z_i$.**
>
> 3. The **estimation of nuisance parameters $\hat{\eta}$ does not affect the asymptotic mean and variance** of the limit distribution of $\hat{\Delta}\_\tau = \frac{1}{n}\sum\_{i=1}^n Z\_i$ (under the assumptions from Remark 4.1). We realized that we missed a formal argument for this step and apologize for this. **We expanded the proof in Appendix A.2 and added a formal argument**. The **intuition is as follows**:
>
>    Remark 4.1. states that this is true for $ \tilde{Y}\_{\hat{\eta}}(x_i) $, i.e., using estimated nuisances $\hat{\eta}$ does not affect the asymptotic mean and variance of $\frac{1}{{n}} \sum_{i=1}^n  \tilde{Y}\_\hat{\eta}(x_i)$. The key fact used here is that we can write the $\frac{1}{n}\sum_{i=1}^n \tilde{Y}\_\hat{\eta}(x\_i) = \underbrace{\frac{1}{n}\sum_{i=1}^n \left( \tilde{Y}\_\hat{\eta}(x\_i) - Y\_\eta(x\_i)\right)}\_{\text{Error due to nuisance estimation}} + \underbrace{\frac{1}{n}\sum\_{i=1}^n Y\_\eta(x\_i) }\_{\text{Oracle nuisance functions}} $. Then the proof in Wager’s book [1] proceeds by showing for the first term that $\sqrt{n}\left(\frac{1}{n}\sum_{i=1}^n \left( \tilde{Y}\_\hat{\eta}(x\_i) - Y\_\eta(x\_i)\right)\right) \rightarrow_p 0$ by using the cross-fitting and rate assumptions on the nuisance estimators. For the second term, we can apply the standard CLT as it only depends on the ground-truth nuisance functions $\eta$.
>
>    **In our case, we proceed similarly** and write, $\hat{\Delta}\_\tau = \frac{1}{n}\sum\_{i=1}^n Z\_i = \underbrace{\frac{1}{n}\sum_{i=1}^n \left( \tilde{Y}\_\hat{\eta}(x\_i) - Y\_\eta(x\_i)\right)}\_{\text{Error due to nuisance estimation}} + \underbrace{\frac{1}{n}\sum\_{i=1}^n \left(Y\_\eta(x\_i) - \hat{\tau}\_2(x\_i)\right)}\_{\text{Oracle nuisance functions}} .$ Note that $\hat{\tau}\_2 (x\_i)$ cancels out in the first term as  $\hat{\tau}\_2 (x\_i)$ does not depend on any nuisance estimator $\hat{\eta}$. Hence, following the same arguments as in proof from Wager’s book, Chapter 2 [1], the first term vanishes and we can apply the standard central limit theorem to the second term.
>
> 4. Steps 1-3 imply that $\hat{\Delta}\_\tau$ is asymptotically normal with population mean $E[Z] = E\_X [ \tilde{Y}\_{\eta}(X) - \hat{\tau}\_2 (X)] = \tau -  E\_X [ \tau\_2 (X)]$ and variance $Var (Z) = Var\_X \left( \tilde{Y}_{\eta} (X) - \hat{\tau}_2(X) \right)$. The population variance can be estimated $\hat{\sigma}\_{\Delta}^2 = \frac{1}{n} \sum\_{i=1}^n \left(\tilde{Y}\_\hat{\eta} (x\_i) -\hat{\tau}\_2(x\_i) -  \hat{\Delta}\_\tau \right)^2$.
>
> **Action**: We **added the missing argument from Step 3 to our new Appendix A.2**.
>
> **Minor**
>
> Thanks for the valuable suggestion.
>
> **Action:** We **corrected the notation and emphasized our use of cross-fitting** in Theorem 4.2. We now explicitly state which component is estimated on which dataset.
>
> **Reference:**
>
> [1] Stefan Wager. Causal Inference: A Statistical Learning Approach, 2024.

---

> ### Comment · Reviewer_sim7 · 2024-11-27
>
> Thank you for your response. My confusion come from the fact that the manner you apply cross-fitting is never explicitly written.  The CLT on $\hat{\Delta}{\tau}$ may hold because $\hat{\eta}$ is built on a data set that does not contain $x_i$. How cross-fitting is applied to your estimators should be clearly mentioned in the text and in the algorithm, so that the implemented procedure is clear to the reader.
>
> Besides, even with cross-fitting, contrary to what you stated, the $Z_i= \tilde{Y}_{\hat{\eta}}(x_i) - \hat{\tau}_2(x_i)$ are not independent as they all depend on the data set used to construct $\hat{\eta}$. Even using cross-fitting with leave-one-out (using all observations but one to build your estimate and compute the quantity of interest on the remaining observation) would not lead to independent $Z_i$. Thus, the proof needs to be clarified.
>
> I believe that the proof of Wager can be extended to your setting, but I would like to see a clear formal proof of this statement.

---

> > ### Author Response · Authors · 2024-11-27
> >
> > Thanks for your further question. We are happy to answer this and replace the informal statement with a formal derivation (see **our revised PDF**).
> >
> > We provide a short summary below. For this, we decompose the rectifier into two parts
> >
> > $$\\hat{\\Delta}\_\\tau \= \\underbrace{\\frac{1}{n}\\sum\_{i=1}^n \\left\[\\tilde{Y}\_{\\hat{\\eta}}(x\_i) \- Y\_{\\eta}(x\_i)\\right\]}\_{\\text{Error due to nuisance estimation}} \+ \\underbrace{\\frac{1}{n}\\sum\_{i=1}^n \\left\[Y\_{\\eta}(x\_i) \- \\hat{\\tau}\_2(x\_i)\\right\]}\_{\\text{Oracle nuisance functions}}.$$
> >
> > The first term denotes **the error introduced by using estimated nuisance functions**. Following Wager’s book, it is  **negligible in probability** if we apply the cross-fitting for estimated nuisance functions and the estimated nuisance functions satisfy the converge rate requirement as we stated in Theorem 4.2. (We provide detailed steps for cross-fitting in our revised **Algorithm 1** and our new supporting lemma from **Appendix A.1**)
> >
> > The second term denotes the mean of differences between pseudo-outcomes, i.e., $Y\_\eta(X)$ based on the oracle nuisance function (**Here, nothing trained on** $\\mathcal{D}^1$ and $\\mathcal{D}^2$) and $\\hat{\\tau}\_2(X)$ which is the CATE estimator trained on $\\mathcal{D}^{2}$ (**particularly, this is independent of** $\mathcal{D}^1$). Thus, $Y\_\\eta(X)$ and $\\hat{\\tau}\_2(X)$ are independent functions, which means that $\mathbf{Y\_\\eta(x\_i) \- \\hat{\\tau}\_2(x\_i)}$ are i.i.d. random variables. Then, the second term follows the CLT.
> >
> > **Actions:**
> >
> > -   We **revised our Algorithm 1** to clarify that we use cross-fitting on $\\mathcal{D}^1$ and sample-splitting on $\\mathcal{D}^2$. Specifically, we added a new statement (in Line 1) where we apply sample splitting (e.g., $\\mathcal{D}^{1}$ is split into $\\mathcal{D}^{1,1}$ and $\\mathcal{D}^{1,2}$). We then explicitly state which of the subsequent lines is trained on which split (e.g., Line 2 is estimated on $\\mathcal{D}^{2,1}$).
> > -   We **replaced our informal statements in our proof in Appendix A.2 with a formal derivation**. For this, we updated Appendix A.2, which now makes use of a new supporting lemma that we provide in our new Appendix A.1.
> >
> > We deeply appreciate your detailed suggestions that helped us to make our statements more formal and make it easier for readers from different backgrounds (e.g., PPI, causal inference, etc.) to follow our work. Thank you!

---

> > > ### Comment · Reviewer_sim7 · 2024-11-28
> > >
> > > Thank you for your detailed response. I am then happy to improve my score.

---

> > > > ### Author Response · Authors · 2024-11-28
> > > >
> > > > Thanks so much for your detailed and positive suggestions and feedback! We will incorporate all action points into our revised version of paper!

---

### Official Review · Reviewer_frUm · 2024-11-03

**Soundness:** 3
**Presentation:** 3
**Contribution:** 2
**Rating:** 6
**Confidence:** 3

**Summary:**

This work presents a method for estimating average treatment effects (ATE) and constructing valid confidence intervals (CIs) from multiple observational datasets. It uses minimal assumptions and leverages prediction-powered inferences for more precise CIs. The approach is validated through theoretical proofs and numerical experiments, with extensions for mixed data sources.

**Strengths:**

The article is well written, and technically sound. The proposed approach is interesting, with great potential for practical application. The previous literature is well integrated, and the paper's novelty is well explicited.

**Weaknesses:**

1. It appears that the novelty of this work is incremental. Use the previously proposed estimator for causal effects and use it in a PPI framework to construct confidence intervals.

2. Lemma 6.1 is a theoretical property of the inverse probability weighting (IPW) estimator and should not be considered a contribution of this paper. Furthermore, the IPW estimator can also be interpreted as an estimation method based on the influence function, while the augmented inverse probability weighting (AIPW) estimator is derived from the effective influence function. Therefore, the statements made in lines 357-360 are inaccurate.

3. The proof of Theorem 5.2 appears to depend on the asymptotic normality of $\tau_2$. However, the $\mathcal{D}_2$ does not seem to satisfy ignorability, suggesting that it is a biased estimator. I have reservations about whether it is indeed asymptotically normal about $\hat{\tau}_2$ and $\tau_2$ at this point.

**Questions:**

This paper is written towards an audience of applied statistics researchers, and is more suited for a statistical journal such as Biometrika. Although the reviewer acknowledges the significance of rigorously discussing the construction of confidence intervals, the methodology presented may not be highly relevant to the broader community of ICLR. For example, it does not discuss neural network or representation learning at all.

---

> ### Author Response · Authors · 2024-11-21
> **Response to Reviewer frUm**
>
> Thank you for your helpful review\! We took all your comments at heart and improved our paper as follows. We thus **uploaded a new PDF** where we highlighted all major changes highlighted in **blue color**.
>
> ### **Response to “Weakness”**
>
> **(W1) Novelty of our method:**
> Thank you for giving us the opportunity to clarify our contributions and how our method is novel. Our main contribution is not the derivation of a new meta-learner but to **propose a novel way to construct valid and more accurate CIs** based on the good asymptotic property of the AIPW estimator in multiple observational datasets and IPW in RCT and observational datasets. Hence, our novelty is the way how we leverage the AIPW to estimate confidence intervals from **multiple observational datasets**. With the help of a large but confounded observational dataset and the idea of prediction-powered inference, we essentially “shrink” the CIs which means offering more precise uncertainty quantifications. As such, **we see our main contributions in Theorem 5.2 and Theorem 6.2**.
>
> Of note, our work is **not** just a simple application of the PPI framework. Rather, the PPI framework does not inform us how to choose the rectifier and how to obtain theoretical properties such as valid CIs. Rather, **this requires us to make new, tailored, and non-trivial derivations to confirm the theoretical properties (as provided by Theorem 5.2 and Theorem 6.2)**. Thereby, we provide new methods for making causal inference from multiple datasets that improve over existing baselines and achieve state-of-the-art performance.
>
> **Action:** We carefully checked our paper and we spelled out our main novelty clearly: we present a novel way using AIPW estimators to construct CIs from multiple datasets with new theoretical guarantees (see **our Theorem 5.2 and Theorem 6.2**).
>
> **(W2) Different of AIPW and IPW:**
> Thanks for your careful comments. We admit that we have made a wrong statement, **which we have now fixed**. The non-centered IF scores are different for IPW and AIPW, for IPW it should be $\\tilde{Y}\_{\\pi}(x\_i) \= \\frac{A\_i Y\_i}{\\pi(x\_i)} \- \\frac{(1-A\_i)Y\_i}{\\pi(x\_i)}$, and for AIPW, it is $\\tilde{Y}\_{\\hat{\\eta}}(x\_i)  \= \\left( \\frac{A}{\\hat{\\pi}(x\_i)} \- \\frac{1-A}{1-\\hat{\\pi}(x\_i)} \\right) Y\_i \- \\frac{A \- \\hat{\\pi}(x\_i)}{\\hat{\\pi}(x\_i) \\left(1-\\hat{\\pi}(x\_i)\\right)} \\left\[ \\left(1-\\hat{\\pi}(x\_i)\\right)\\hat{\\mu}\_1(x\_i) \+ \\hat{\\pi}(x\_i)\\hat{\\mu}\_0(x\_i)\\right\]$. To clarify, the key difference between **Sections 5 & 6** in our main paper is that we do not need to estimate the propensity score in the RCT dataset and only AIPW holds asymptotic normality in the observational dataset.
>
> **Action:** We corrected the statement in our revised paper.
>
> **(W3) Asymptotically of $\\hat{\\tau}\_2$:**
> Thank you for this suggestion. We do not rely on the asymptotic normality of $\\tau\_2$, as it directly follows from the central limit theorem (CLT). Specifically, given that $\\tau\_2$ is derived from a sum of independent random variables, the CLT ensures its asymptotic normality under standard regularity conditions, i.e., $\\tau\_2 \\overset{d}{\\to} \\mathcal{N}(\\mu\_2, \\sigma\_2^2), \\text{ as } n \\to \\infty$, where $\\mu\_2$ and $\\sigma\_2^2$ denote the mean and variance of $\\tau\_2$, respectively. The key focus of our analysis lies in the asymptotic normality of $\\tau\_1$. In the observational confounded dataset, only when we apply the average non-centered IF scores from the AIPW as the estimation of ATE the $\\hat{\\tau}\_2$ has the asymptotic normality to the oracle ATE in the population. After that, we can state: $\\tau\_1 \\overset{d}{\\to} \\mathcal{N}(\\mu\_1, \\sigma\_1^2), \\text{ as } n \\to \\infty$, where $\\mu\_1$ and $\\sigma\_1^2$ represent the mean and variance of $\\tau\_1$.
>
> **Action:** We added a new section to explain the asymptotic properties of $\tau\_2$ (see **our new Appendix A.3**).
>
> ### **Response to “Questions”**
>
> Thank you for your feedback. A particular strength of **our method is that it is flexible** and that **it can be used with various base learners including neural networks or other representation learning methods**. Upon reading your question, we realized that we show the applicability of our method also to neural learning, and we thus performed new experiments where we demonstrate when our method is applied on top of neural networks. Again, we find that our proposed method is highly effective.
>
> **Action:** We **performed new experiments with neural networks** to show how our method makes contributions to representation learning (see **our new Appendix G.1**).

---

> > ### Comment · Reviewer_frUm · 2024-11-27
> >
> > The reviewer acknowledged that the concerns are addressed and increased the score accordingly.

---

> > > ### Author Response · Authors · 2024-11-27
> > >
> > > Thank you for your positive feedback! We will incorporate all action points into our revised version of paper.

---

### Official Review · Reviewer_k7Su · 2024-11-03

**Soundness:** 3
**Presentation:** 3
**Contribution:** 3
**Rating:** 8
**Confidence:** 3

**Summary:**

This paper considers the problem of estimating confidence intervals for average treatment effects when given access to a small, unconfounded observational dataset and a larger, confounded observational dataset. Specifically, the problem setting assumes that the covariates for both datasets come from the same population, but the propensity score may differ and the potential outcomes are only independent of treatment assignment in the smaller unconfounded dataset.

The paper's method builds on the prediction-powered inference (PPI) framework, which creates an estimate of the ATE via the confounded dataset and then uses the unconfounded dataset to adjust it and build the appropriate confidence intervals. The paper proves that their method asymptotically results in valid confidence intervals. The method is evaluated on both synthetic and real data, showing its gains over using only the unconfounded dataset.

**Strengths:**

The paper addresses an important issue that has not been covered in the literature before. The paper also does a good job of placing itself within the literature. The paper also does a good job of precisely backing up their claims with theorems.

**Weaknesses:**

On the theoretical side, the paper only shows the asymptotic validity of the proposed method. There are no results demonstrating the width of the resulting confidence intervals. One would like to know under what circumstances the given method will produce a confidence interval that converges on exactly 1-$\alpha$.

Moreover, the paper does not give any theoretical insight into when it is beneficial to use both datasets as opposed to just the unconfounded dataset. There should be some result showing that the confidence intervals are tighter for proposed method than for the naive method.

**Questions:**

Under what circumstances can it be proved that the proposed method is actually better than operating on just the unconfounded dataset?

---

> ### Author Response · Authors · 2024-11-21
> **Response to Reviewer k7Su**
>
> Thank you for your positive evaluation of our paper\! We took all your comments at heart and improved our paper accordingly. We thus **uploaded a new PDF** where we highlighted all major changes highlighted in **blue color**.
>
> ### **Response to “Weakness”**
>
> **(W1) Comparison of the width of CIs and coverages:**
> Thank you for your thorough suggestions about the comparison of the width of resulting CIs. We apologize for the unclear in our main paper. To clarify, the right subfigures in **Figures 4 & 5 in the main paper** illustrate the comparison of the widths of CIs. From those figures, we show that the widths of proposed CIs are significantly shorter than the naive method. This **shows that our method successfully shrinks the width of CIs as desired**. Also, following our proof in **Appendix A.2,** we show that **our method provides valid $1-\\alpha\\%$ confidence intervals**.
>
> **Action:** We improved our descriptions of Figures 4 and 5 to explain that these show the width and that our method shrinks the CI width as desired.
>
> **(W2) Comparison to the naive method:**
> Thank you for your detailed comments. To clarify, the yellow line in the experimental plots represents the width of the confidence intervals produced by the naive method, i.e., $\\hat{\\tau}^{\\mathrm{AIPW}}$, ($\\mathcal{D}^1$ only). Note that the naive method is exactly what you suggested, namely, using the unconfounded dataset only. Hence, our plots allow us to empirically validate that our method is **better** than the naive way of using only the unconfounded dataset (see the yellow lines in Figures 4, 5, etc.).
>
> We further explain theoretically why our method is better than your suggested way of using only the unconfounded dataset. We added a new paragraph to our paper where we answer the question “Why is our method better than using the unconfounded dataset only?” (see our response to your question below)
>
> Finally, we realized upon reading your question that we could offer a more comprehensive evaluation of our method under different levels of confounding of $\\mathcal{D}^1$. This allows us to better understand the source of gain. Hence, we performed new experiments for this (see **our new Appendix G.5**). First, our method is robust and performs best in all settings. Second, we see that the source of gain for our method is larger for settings with high confounding, which can be expected.
>
> **Action:** We clarified that the naive method in our paper matches your proposed idea of using only the unconfounded data and making it clearer in the revised paper. We added new experiments with different levels of confoundedness (see our **new** **Appendix G.5)**
>
> ### **Response to “Questions”**
>
> Thank you. We added a new paragraph to our paper where we answer the question “Why is our method better than using the unconfounded dataset only?” Indeed, we can even show that our method is almost always better. The reason is the following. As the measure of fit $\\hat{\\tau}\_2$ is sufficiently accurate, the rectifier is almost equal to zero, i.e., $\\hat{\\Delta} \\approx 0$. Then, the variance of the rectifier is significantly smaller than the variance of estimated non-centered IF scores, $\\hat{\\sigma}\_{\\Delta}^2 \\leq \\hat{\\sigma}_{\tau_2}^2$. Given the large size of $\\mathcal{D}\_2$, the variance of the estimated conditional treatment effect goes to zero, since the estimated variance should be divided by the sample size of $\\mathcal{D}\_2$, i.e., $N$. As a result, the variance (and thus the CI width) is smaller when using our method than when using only the unconfounded dataset, which highlights the strengths of our proposed method from a theoretical perspective.
>
> **Action:** We **added the above explanation to our paper** as part of a new paragraph “Why is our method better than using the unconfounded dataset only?” (see our new elaborations in our **revised Section 5\)**.

---

> > ### Comment · Reviewer_k7Su · 2024-11-26
> >
> > Thank you for your response and addressing my concerns. I have updated my score.

---

> > > ### Author Response · Authors · 2024-11-26
> > >
> > > Thank you for your positive feedback! We will incorporate all action points into our revised version of paper.

---

### Author Response · Authors · 2024-11-21
**Response to all reviewers**

Thank you very much for the constructive evaluation of our paper and your helpful comments\! We addressed all of them in the comments below. We also uploaded our **revised version of the paper**.

Our main **improvements** are the following:

* **Additional theoretical analysis:** We added a very detailed and technical illustration in our **new Appendix H** showing the key differences between our method and the work from van der Laan et al., 2024\. The method by van der Laan et al involves the matrix inversion problem, which is one of the key steps while reweighting the canonical gradient of the beta-component which is important since ATMLE needs the canonical gradient to iteratively estimate the targeted estimand. This is the reason why their method is unstable and often breaks. We provide further details about the technical problems of the method as well as an in-depth comparison in our **new Appendix H**.
* **Additional experimental results** We expanded our experimental results in our **new** **Appendix G.** Specifically, we performed the following new experiments:
  * **New neural instantiations of our method:** We instantiated our method with neural networks. Thereby, we show how our method contributes to representation learning and demonstrate that our method can effectively handle neural base methods (see our **new** **Appendix G.1**).
  * **New base learners:** We replaced the regression model for nuisance parameters from linear regression to XGBoost  (see our **new** **Appendix G.2**). Thereby, we show the flexibility of our method to other base learners.
  * **High-dimensional covariates:** We generated a synthetic dataset with high-dimensional covariates. This allows us to study how our method behaves in settings with increasing complexity (see **our new Appendix G.3**).
  * **Strength of dependence:** We generated an additional covariate as the mean of all other covariates to construct the collinearity in input space. That is, we here examine how our method when the dependence on the covariates varies (see our **new** **Appendix G.4**).
  * **Different strengths of (un)confounding in $\\mathcal{D}\_1$:** We relaxed the “Unconfoundedness” assumption with three different confounding scenarios while fixing the confounding in $\\mathcal{D}\_2$. Thereby, we can study the performance of our method in settings with different strengths of (un)confounding (see our **new** **Appendix G.5**).
  * **AIPW to RCT+obs datasets and A-TMLE to observational datasets:** We apply the AIPW to the RCT dataset (see our **new** **Appendix G.6**) and A-TMLE to the observational datasets (see our **new** **Appendix G.7**). Thereby, we show the robustness of our method to RCT and observational datasets.
  * **Increasing sample size in $\\mathcal{D}\_1$:** We varied the size of the small dataset $\\mathcal{D}\_1$ from 100 to 2500\. Thereby, we further assess the role of the size of $\\mathcal{D}\_1$ (see our **new** **Appendix G.8**).

  We find that **our method is always better than the na{\\”i}ve method under all settings**. This confirms our theoretical contributions and verifies the effectiveness of our method.

We incorporated all changes (indicated with the label **Action** in our individual responses) into the revised version of our paper and highlighted all key changes in **blue color**. Given these improvements, we are confident that our paper provides valuable contributions to the causal machine learning literature and is a good fit for ICLR 2025\.

---

### Meta-Review · Area_Chair_hRUD · 2024-12-19

**Metareview:**

**Summary**: This paper proposes a new method to generate reliable confidence intervals for average treatment effects using multiple datasets where additional data set samples may have residual unobserved confounding. Authors build on the prediction-powered inference (PPI) framework to estimate ATE on confounded data, rectifying the bias using the smaller unconfounded data, and using sample statistics of the bias for generating confidence intervals. The procedure uses CATE estimates from the larger dataset to obtain a DR CATE estimate using sample splitting, followed by rectifying using the efficient influence function and subsequently deriving confidence intervals from the summary statistics. The framework is flexible by allowing ML methods used for nuisance estimation in high dimensions and applicable to AIPW, IPW estimators. Theoretical results show asymptotic coverage of the confidence intervals.

**Strengths**:
1. Reliable uncertainty quantification of ATE is a critical problem for improving causal inference.
2. The paper further attempts to tighten interval estimates by leveraging potentially confounded data thereby leveraging the power of additional samples. The contribution is significant and novelty is clear and easy to follow.
3. Theoretical claims while straightforward are well presented and justified.
4. Empirical evaluation clearly shows better coverage compared to prior work.

**Weaknesses**:
1. Multiple reviewers pointed out that the empirical evaluation is relatively weak for the standards of ICLR, but the focus of the contribution
2. Considering that the contribution is more statistical, i.e., targeted toward better inference without heavy focus on the representation learning of the nuisances, fit to ICLR was not clear for multiple reviewers

**Justification**: Overall all reviewers agree that the contribution is valuable. Multiple reviewers had clarifying questions on the details. There were many clarity issues in the writing in the initial version which were subsequently addressed, proof statements clarified, additional empirical evaluation added, including updates to the written text to add pertinent details e.g., on use of sample splitting. Overall after multiple iterations of discussions between reviewers and authors, it is clear that clarity issues where addressable and not fundamental issues with the contribution. Overall the novelty of the contribution is explicit, demonstration of empirical benefit clear, and multiple reviewers raised their scores post rebuttal. Considering the novelty of the contribution, the quality of the writing, and potential interest to a small but significant subset of ICLR readers, I am recommending an accept.

**Additional Comments On Reviewer Discussion:**

No major concerns were brought up during discussion.

---

### Decision · Program_Chairs · 2025-01-22

Accept (Poster)